# Application of the shipborne remote sensing supersite OCEANET for profiling of Arctic aerosols and clouds during Polarstern cruise PS106

Hannes J. Griesche, Patric Seifert, Albert Ansmann, Holger Baars, Carola Barrientos Velasco, Johannes Bühl, Ronny Engelmann, Martin Radenz, and Yin Zhenping

Leibniz Institute for Tropospheric Research (TROPOS), Leipzig, Germany

**Correspondence:** Hannes Jascha Griesche (griesche@tropos.de)

**Abstract.**

From 25 May to 21 July 2017, the research vessel Polarstern performed the cruise PS106 to the high Arctic in the region north and northeast of Svalbard. The mobile remote sensing platform OCEANET was deployed aboard Polarstern. Within a single container, OCEANET houses state-of-the-art remote sensing equipment, including a multi-wavelength Raman polarization lidar Polly[XT] and a 14-channel microwave radiometer HATPRO. For the cruise PS106 the measurements were supplemented by a motion-stabilized 35-GHz cloud radar Mira-35. This article describes the treatment of technical challenges which were immanent during the deployment of OCEANET in the high Arctic. This includes the description of the motion stabilization of the cloud radar Mira-35 to ensure vertical-stare observations aboard the moving Polarstern as well as the applied correction of the vessels heave rate to provide valid Doppler velocities. The correction ensured a leveling accuracy of $\pm 0.5°$ during transits through the ice and a performed ice floe camp. The applied heave correction reduced the signal induced by the vertical movement of the cloud radar in the PSD of the Doppler velocity by a factor of 15. Low-level clouds in addition frequently prevented a continuous analysis of cloud conditions from synergies of lidar and radar within Cloudnet, because the technically determined lowest detection height of Mira-35 was 165 m above sea level. To overcome this obstacle, an approach for identification of the cloud presence solely based on data from the near-field receiver of Polly[XT] at heights from 50 m and 165 m above sea level is presented. We found low level stratus clouds, which were below the lowest detection range of most automatic ground-based remote sensing instruments during $25\%$ of the observation time.

We present case studies of aerosol and cloud studies to introduce the capabilities of the data set. In addition, new approaches for ice crystal effective radius and eddy dissipation rates from cloud radar measurements and the retrieval of aerosol optical and microphysical properties from the observations of Polly[XT] are introduced.

## 1 Introduction

The Arctic is one of the hot spots of global climate change. This is observed as a change of several parameters such as the drastic decline of the Arctic sea ice during all seasons, but especially in summer, in both extend and thickness (Meier et al., 2014). Also, in the past 30 years the mean Arctic near surface air-temperature anomaly increased at least by a factor of two faster

compared to the global mean (Serreze and Barry, 2011). These phenomena, summarized by the term Arctic Amplification,
are assumed to be due to several feedback mechanisms, e.g., the surface albedo feedback, lapse rate feedback, a change in the meridional atmospheric and oceanic mass and energy transport pattern, and an alteration in cloud cover, aerosol occurrence, and atmospheric moisture content (Wendisch et al., 2017). However, there is still a lack of understanding in the interplay of these feedback mechanisms as well as in quantifying their relative importance and magnitude (Serreze and Barry, 2011; Pithan and Mauritsen, 2014; Goosse et al., 2018).

The radiative effect of clouds is a major source of uncertainty in this matter. Arctic clouds have a high variability in their radiative effects and in their impact on the surface energy balance (Yeo et al., 2018). The cloud-related radiative impacts have been found to be both positive (i.e., clouds have a warming effect) as well as negative (Goosse et al., 2018). A higher fraction of low clouds, e.g. caused by a decreased sea ice extent, increases the downwelling longwave radiation during polar night and thus induces a positive feedback. A higher fraction of liquid water in mixed-phase clouds due to a warmer climate, on the other
hand was found to increase the cloud albedo. This in turn enhances the reflection of incoming shortwave radiation at the top of the atmosphere during polar day (Goosse et al., 2018) and thus produces a negative feedback. Yet the underlying processes controlling Arctic cloud phase and occurrence, and hence the connected feedback mechanisms driving Arctic Amplification are not completely understood (e.g., Shupe et al., 2013; Kalesse et al., 2016a).

Though being a key requirement to study the Arctic energy budget, detailed observations of Arctic clouds still are rare.
Measurements from ground based stations, such as the observatories of the International Arctic Systems for Observing the Atmosphere (Uttal et al., 2016) are of great value, e.g. due to their possibilities of conducting long-term observations. However they are limited to their location and influenced by their surrounding orography. Drifting buoys on the other hand can enter any place in the Arctic ice and thus are very valuable in this harsh environment. Their equipment gets increasingly sophisticated and since a few years buoys equipped with autonomous lidar systems (Mariage et al., 2017) have been in use, giving
them the opportunity to measure vertical profiles of the atmospheric column. But they still are limited in their payload and can not yet replace measurements from observatories or campaigns. To study Arctic clouds, different aircraft campaigns have been conducted in recent years (e.g., Curry et al., 2000; Jacob et al., 2010; McFarquhar et al., 2011; Wendisch et al., 2019). While airborne measurements yield an unique, accurate description of the observed cloud, they lack the ability to measure continuously the entire tropospheric column over a long period, a feature active remote sensing observations can offer. Given
this capability, ground-based remote sensing observations are suitable to investigate the spatio-temporal distribution of clouds (Bühl et al., 2013), their phase partitioning (de Boer et al., 2009; Zhang et al., 2014; Kalesse et al., 2016a), and their interaction with aerosols (Seifert et al., 2010). These data sets serve, e.g., as basis for model evaluation (Illingworth et al., 2007; Neggers, 2019) and radiative transfer calculations (Ebell et al., 2019; Barrientos Velasco et al., 2020). Hence, additionally to the airborne campaigns, several shipborne campaigns equipped with remote sensing instrumentation have been conducted in
the past years in the Arctic (e.g., Uttal et al., 2002; Tjernström et al., 2004; Tjernström et al., 2014; Granskog et al., 2018; Wendisch et al., 2019). Observations of space-borne cloud radar and lidar, as done aboard Cloudsat (Stephens et al., 2008) and CALIPSO (Winker et al., 2010) provide in addition a large-scale overview of the Arctic cloud coverage (Liu et al., 2012). But the respective data sets lack information about the lowest cloud levels. Nevertheless, there are still only a few studies of

sea-motion-stabilized cloud radars, whose availability is a necessary requirement to determine also cloud vertical dynamics accurately from a shipborne platform.

In order to study the feedback mechanisms causing Arctic Amplification, the initiative **A**rcti**C** **A**mplification: **C**limate Relevant Atmospheric and Surfa**C**e Processes and Feedback Mechanisms (AC)[3] conducted two complementary field campaigns in the Arctic summer of 2017: **A**rctic **CL**oud **O**bservations **U**sing airborne measurements during polar **D**ay (ACLOUD), an airborne campaign performed with the research aircraft Polar 5 and Polar 6, and the **P**hysical feedbacks of **A**rctic boundary layer, **S**ea ice, **C**loud and **A**eroso**L** (PASCAL) expedition deployed on and around the research ice breaker Polarstern (Macke and Flores, 2018; Wendisch et al., 2019). These campaigns took place in May and June 2017 in the regions north and northeast of Svalbard with the aim to combine remote sensing and in-situ observations. During PASCAL, a two-week ice floe camp was performed in the vicinity of Polarstern and a large number of auxiliary measurements were conducted on the ice. PASCAL was the first part of the split Polarstern cruise PS106 which took place from 25 May until 21 July 2017. During the whole PS106 cruise, measurements with the multiwavelength polarization lidar Polly[XT]_OCEANET, a 35-GHz cloud radar Mira-35 and a microwave radiometer HATPRO (**H**umidity **A**nd **T**emperature **PRO**filer) of the OCEANET platform were conducted aboard Polarstern. Within (AC)[3] the OCEANET observations have the essential role to describe the temporal evolution of the vertical structure of aerosol and clouds in the central Arctic. They constitute the prerequisite for further studies of aerosol-cloud interaction, model evaluation or radiative transfer modeling, which are partly covered by other subprojects of (AC)[3]. Scope of this study is thus to introduce the instrumentation and data analysis methods which are used to produce the OCEANET-based cloud and aerosol data sets for the cruise PS106.

In Section 2 of this paper, a detailed description of the OCEANET instruments and the auxiliary observations is given. The applied motion stabilization and heave rate correction of the cloud radar Doppler velocity, the data processing based on the synergistic Cloudnet algorithm (Illingworth et al., 2007) and the development of auxiliary retrievals for processing within Cloudnet are described in Sect. 3. In Section 4 the products derived from the OCEANET measurements using Cloudnet are illustrated by means of different case studies from the time period of the ice flow camp. The potential of Polly[XT]_OCEANET to characterize the free-tropospheric aerosol is also highlighted. In addition, a statistical overview about the observed cloud vertical structure with a special focus on low-level clouds during PS106 is presented. A final summary and conclusions are given in Sect. 5.

## 2 Instrumentation

During the complete PS106 campaign in the central Arctic in summer 2017 (see Fig. 1) a comprehensive number of remote sensing instruments was deployed aboard the research vessel (RV) Polarstern to conduct continuous observations of clouds and aerosols. To a large extent, these instruments were comprised in the OCEANET-Atmosphere observatory (Kanitz et al., 2013a). Additionally, auxiliary instruments for in-situ and remote sensing observations installed aboard Polarstern as well as during a two-week ice floe-camp, which was performed in the vicinity of the RV, were utilized for the studies presented in

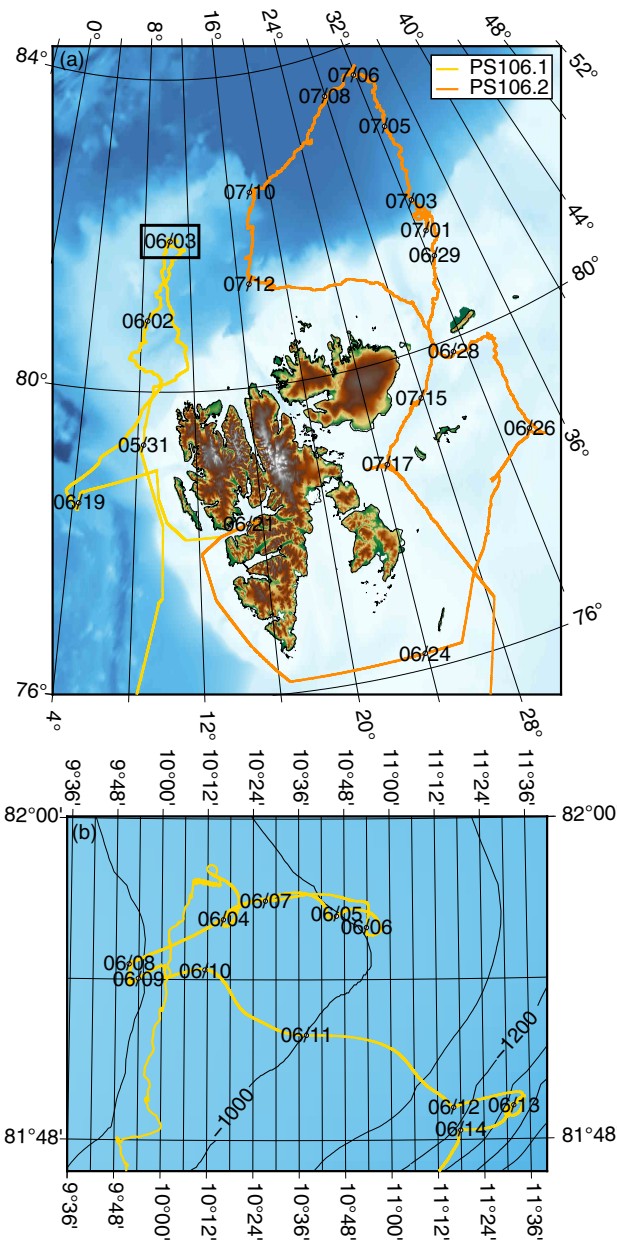

**Figure 1.** Track of RV Polarstern during PS106 (a). PASCAL (PS106.1, yellow) was the first part of PS106 and was accompanied by a two-week drift (b) during which the ice floe camp was performed. Map created with GMT (Wessel et al., 2019).

here. The first part of PS106 (PS106.1 / PASCAL) was accompanied by the ACLOUD aircraft campaign (Ehrlich et al., 2019) about both of which a brief overview is given by Wendisch et al. (2019).

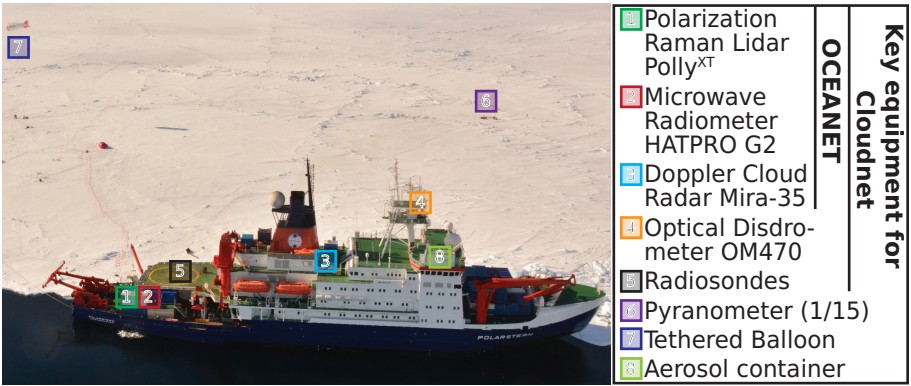

**Figure 2.** Polarstern during the ice floe camp performed during the PASCAL campaign. Annotated are the locations of some selected instruments for atmospheric measurements. (1–5) indicate the positions of the key instruments used for Cloudnet processing: Polly$^{XT}$, HATPRO, Mira-35, disdrometer and radiosondes. (6) denotes one of the 15 pyranometers comprising pyranometer network, at (7) was the tethered balloon launching site and at (8) aerosol in-situ measurements had been conducted. (1–3) are permanent part of OCEANET. Picture by N. Fuchs.

The location of the OCEANET equipment and of the auxiliary Polarstern instruments deployed during PS106 that were used within this study are depicted in Fig. 2. Table 1 summarizes the technical details of the key equipment applied in the synergistic Cloudnet processing that is further described in Sect. 3.3. In the following the different instruments will be briefly introduced.

## 2.1 OCEANET

The OCEANET-Atmosphere observatory was already frequently operated aboard Polarstern (Kanitz et al., 2013b; Bohlmann et al., 2018; Yin et al., 2019). Yet, so far only for the transects from the northern to the southern hemisphere (or vice versa) but never in the Polar regions. Its container is by default equipped with the multi-wavelength polarization Raman lidar Polly$^{XT}$_OCEANET (hereafter referred to as Polly$^{XT}$), to provide continuous profiles of cloud and aerosol properties (Engelmann et al., 2016). Additionally, a 14-channel microwave radiometer (MWR) HATPRO (Rose et al., 2005) for measurements of column-integrated liquid water and water vapor content and profiles of atmospheric temperature, a standard meteorological station, a pyranometer and a pyrgeometer for incoming short- and longwave radiation observations, as well as an all-sky camera for passive visible observations of the full sky were installed. During PS106, OCEANET was complemented for the first time with a vertically-pointing motion-stabilized 35-GHz polarimetric Doppler cloud radar of type Mira-35 (Görsdorf et al., 2015) for continuous vertically resolved measurements of Doppler spectra produced by cloud vertical motions.

The Polly$^{XT}$ system measures profiles of particle backscatter coefficient at three wavelengths (355, 532 and 1064 nm), and of extinction coefficient as well as of the linear depolarization ratio at two wavelengths (355 and 532 nm), respectively , details see Baars et al. (2016). Four near-field channels for detection of elastically and Raman-scattered light from nitrogen molecules are implemented at 355, 387, 532 and 607 nm to enable observations already at low heights starting at about 50 m above the

instrument. An additional channel for detection of Raman-scattered light from water vapor at 407 nm allows the retrieval of the water vapor mixing ratio (Dai et al., 2018) during low sunlight conditions. From the Polly[XT] backscatter and extinction measurements, aerosol classification by their optical properties (Müller et al., 2007; Baars et al., 2017) up to the retrieval of particle size distribution and number concentration (Müller et al., 1999; Baars et al., 2012) can potentially be derived. The

polarization-sensitive detection channels allow to distinguish between spherical and non-spherical aerosol and cloud particles (Kanitz et al., 2013a) and, for instance, to separate dust and non-dust particles in mixed aerosol layers (Baars et al., 2011). By applying the shape-detection capabilities of the polarization channels for the discrimination of spherical liquid droplets from non-spherical ice particles, heterogeneous ice formation in mixed-phase clouds can be studied (Seifert et al., 2015). Another application of depolarization observations in mixed-phase cloud studies is the estimation of cloud condensation nuclei (CCN)

and ice nucleating particle (INP) concentrations (Mamouri and Ansmann, 2016). Due to the relatively short wavelengths of the lidar, e.g. compared to the cloud radar, it follows that the lidar is sensitive to rather small particles such as aerosols or small cloud droplets. Also attenuation, especially due to liquid clouds, has to be considered.

The MWR HATPRO provides estimates of the liquid water path (LWP), integrated water vapor (IWV), as well as humidity and temperature profiles with a temporal resolution of 1 Hz. The MWR measures the emission of radiation from the atmosphere

in two frequency bands ranging from 22.24 – 31.4 GHz and from 51.0 – 58.0 GHz at 14 different channels. The MWR data sets shown in this study are based on a retrieval that was created based on a long-term radiosonde data set from Ny-Ålesund, Svalbard, Norway (78.9° N, 11.8° E, 11 m hasl, WMOCode 6260) according to Löhnert and Crewell (2003).

During PS106, the Ka-band Doppler radar Mira-35 was set-up to emit pulses with a width of 208 ns at a pulse repetition frequency of 5000 Hz. This corresponds to a vertical resolution of 31.18 m. The upper limit of the measurement range was set to

15 km. The Doppler spectrum was derived from the backscattered signals of 256 consecutive pulses. To allow the correction of the cloud radar data for the vessel movement, the whole spectrum (including noise) has been stored with a temporal resolution of 4 Hz and a Doppler resolution of $0.08 \, \mathrm{m \, s^{-1}}$. This correction has been done in a post processing procedure which is explained in Sect. 3.1. From the profiles of the Doppler spectra, the different Doppler moments such as radar reflectivity, Doppler velocity, and Doppler spectral width were determined as described in Görsdorf et al. (2015). The linear depolarization ratio (LDR)

was obtained from the ratio of the radar reflectivity factor observed in the co- and cross-channels of Mira-35 and provides information about the hydrometeor shape (Bühl et al., 2016). In contrast to the lidar, the longer wavelength of operation of the cloud radar defines its sensitivity to range from cloud hydrometeors to slight precipitation. Especially in the case of shallow stratiform clouds, as they dominated the measurements during PS106, attenuation effects can be neglected. The OCEANET data sets of HATPRO, Polly[XT] and Mira-35 are publicly available through the Open Access library PANGAEA (Griesche et al.,

2020b, c, 2019).

## 2.2 Auxiliary instrumentation

Added value of the OCEANET measurements can be obtained when they are accompanied by additional observations. During the two-week ice floe camp performed in the frame of PASCAL, a tethered balloon site was set up for turbulence and radiation observations (Egerer et al., 2019) and a network covering 15 pyranometers to determine the spatial variability of the solar

radiation was installed (Barrientos Velasco et al., 2020). In the context of this study, the turbulence as determined from the three-dimensional wind vector measured with high temporal resolution of several tens of Hertz by an ultrasonic anemometer attached to the tether of the balloon was used. To obtain mass and number concentration as well as optical properties and filter samples of the aerosol at the surface, a container equipped with instrumentation for aerosol in-situ measurements was installed on the deck of Polarstern and was measuring continuously during the whole two-month cruise (Kecorius et al., 2019).

Also aboard Polarstern, measurements of the optical thickness of the cloud-free atmosphere were performed using a hand-held Solar Light Microtops Sun photometer. The Sun photometer measurements are already available through the **AE**rosol **RO**botic **NET**work (AERONET) project. An optical disdrometer, which is part of the OceanRAIN network (Klepp et al., 2018), mounted on the crows nest of the RV was continuously measuring the precipitation rate for different hydrometeor types and size bins. Additionally, launches of Vaisala RS92-SGP radiosondes (Jensen et al., 2016) were conducted every 6 hours

(shortly before 5, 11, 17 and 23 UTC to reach 100 hPa approximately at 6, 12, 18 and 24 UTC) to obtain in-situ profiles of temperature ($\Delta$T=0.5°C), relative humidity ($\Delta$RH=5%), pressure ($\Delta$p=1 hPa), and horizontal wind speed ($\Delta$v=0.15 ms$^{-1}$) and direction ($\Delta$°=2°).

## 3   Data processing and synergistic retrievals

Aim of the OCEANET observations from PS106 was to provide a continuous vertically-resolved view on cloud and aerosol macro- and microphysical properties in order to enhance the understanding of the Arctic atmosphere system and to support partner projects with data sets for radiative transfer calculations and turbulence studies. To derive continuous products of cloud and aerosol properties, the shipborne OCEANET remote sensing observations were processed using the synergistic retrieval algorithm Cloudnet (Illingworth et al., 2007). In this section, we describe the extension of the standard Cloudnet algorithms

by additional simple but operationally applicable products providing estimates of cloud droplet and ice crystal effective radius and the cloud-turbulence parameter eddy dissipation rate (EDR). The procedure for minimizing the influence of the RV motion on the measurement of vertical velocities with Mira-35, which are required for the EDR retrieval, is also explained below.

### 3.1   Correction of vertical-stare cloud radar observations for ship motion

A structural requirement to derive valid vertical velocity from a Doppler cloud radar is a vertical pointing radar without an
own vertical-velocity component. When the cloud radar is pointing off-zenith, the measured vertical-stare Doppler velocity will be biased by an additional component introduced by the horizontal wind. Based on high resolved horizontal wind data and the radar beam incident angle, a correction is possible for this bias (Wulfmeyer and Janjić, 2005). For PS106, a different approach was chosen. Similar to the approach described by Achtert et al. (2015), the cloud radar was mounted on an active stabilization platform (Fig. 3 (a)), which was in our case a predecessor of the SOMAG AG Jena – GSM 4000 (SOMAG, 2017).
This platform actively leveled out the roll and pitch movement of the RV, ideally in a way that no correction of horizontal-wind effects was necessary.

**Table 1.** Overview of the instrumentation deployed during PS106 that had been used for processing of the OCEANET observations.

| Instrument<br>Type | Reference | Measured Quantities | $\nu$: Frequency<br>$\lambda$: Wavelength<br>R: Range of Measurement<br>P: Precision | Time Resolution |
|---|---|---|---|---|
| **Raman Lidar** | | | | |
| Polly[XT] | Engelmann et al. (2016) | Particle backscatter coefficient | $\lambda = 355, 532, 1064\,\mathrm{nm}$<br>R: 0.1–15 km, 0–1 $\mathrm{km^{-1}\,sr^{-1}}$<br>P: 7.5 m; 1e-5 $\mathrm{km^{-1}\,sr^{-1}}$ | 10 min -1 hour |
| | | Particle extinction coefficient | $\lambda = 355, 532\,\mathrm{nm}$<br>R: 0.3–5 km, 0–10 $\mathrm{km^{-1}}$<br>P: 300 m; 1e-2 $\mathrm{km^{-1}}$ | |
| | | Particle linear depolarization ratio | $\lambda = 355, 532\,\mathrm{nm}$<br>R: 0.1–15 km, 0–0.5;<br>P: 7.5 m; 0.02 | |
| **Microwave Radiometer** | | | | |
| RPG HATPRO-G2<br>first generation<br>dual profiler | Rose et al. (2005) | Integrated water vapor (IWV) | $\nu = 22.24$–$31.4\,\mathrm{GHz}$<br>R: 0–35 $\mathrm{kg\,m^{-2}}$<br>P: 0.2 $\mathrm{kg\,m^{-2}}$ | 1 Hz |
| | | Liquid water path (LWP) | $\nu = 22.24$–$31.4\,\mathrm{GHz}$<br>R: 0–1 $\mathrm{kg\,m^{-2}}$<br>P: 0.02 $\mathrm{kg\,m^{-2}}$ | |
| | | Brightness temperature (TB) | $\nu = 51.0$–$58.0\,\mathrm{GHz}$<br>R: 0–330 K<br>P: 0.2–1 K | |
| **Doppler Cloud Radar** | | | | |
| Metek Mira-35 | Görsdorf et al. (2015) | | $\nu = 35.5\,\mathrm{GHz}$ | 3.5 sec |
| | | Radar reflectivity factor | R: 150–13000 m; -55–20 dBZ<br>P: 3 m; 2 dBZ | |
| | | Linear depolarization ratio | R: 150–13000 m; -26-0 dB<br>P: 30 m; 1 dB | |
| | | Hydrometeor vertical velocity | R: 150–13000 m; -11–11 $\mathrm{m\,s^{-1}}$<br>P: 30 m; 0.08 $\mathrm{m\,s^{-1}}$ | |
| **Optical Disdrometer** | | | | |
| Eigenbrot ODM470 | Klepp et al. (2018) | Particle size distribution | $\lambda = 880\,\mathrm{nm}$<br>R: 0.04–22 mm<br>P: 0.03–0.5 mm | 1 min |
| **Tethered balloon** | | | | |
| Ultrasonic anemometer | Egerer et al. (2019) | 3-D wind vector | R: 0–20 $\mathrm{ms^{-1}}$ | 50 Hz |
| Metek uSonic-3 Class A | | sonic (virutelle) temperature | R: -35–55$^\circ C$ | 50 Hz |
| **Pyranometer network** | | | | |
| 15 Pyranometer<br>EKO Instruments<br>ML-020VM | Barrientos Velasco et al. (2020) | Spectral irradiance | R: 0.3–1.1 $\mu m$ | 1 Hz |
| **Sunphotometer** | | | | |
| Solar Light<br>540 Microtops II | Porter et al. (2001) | Aerosol optical thickness | $\lambda = 340, 380, 440,$<br>$500, 675, 870, 936, 1020\,\mathrm{nm}$ | On demand |

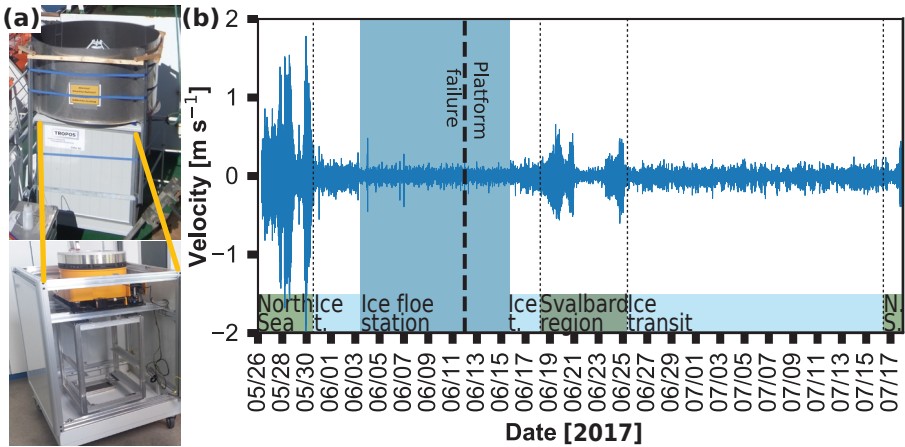

**Figure 3.** (a) The cloud radar aboard Polarstern and the stabilization platform. (b) shows the time series of the cloud radar heave rate during PS106. The thick dashed vertical line indicates the moment when the stabilization platform got a malfunction. At the bottom a rough localization of Polarstern is annotated (green: North Sea (N.S.), light blue: Ice transit (Ice t.), dark blue: Ice floe camp, dark green: Svalbard region).

Figure 4 shows a comparison of the pitch and roll angle time series during ice breaking conditions from 1 June 2017 07:00 UTC – 3 June 2017 8:00 UTC measured by the vessel's own inertial measurement unit (IMU) and directly at the cloud radar. As the platform itself did not serve any position determination we made use of a single board computer (Beaglebone Blue) with integrated IMU. During the ice transit and the ice floe camp periods, the stabilization platform ensured an accuracy of the leveling of $\pm 0.5°$. The 2-sigma standard deviation during the ice transit (ice floe camp) was found to be $0.32°$ $(0.34°)$, thus 95% of the datapoints show an accuracy of $89.68°$ $(89.66°)$. During the open-sea passage of RV Polarstern, the accuracy of the stabilization was reduced to around $\pm 1°$ with a 2-sigma standard deviation of $0.7°$.

An additional bias of the true Doppler velocity can occur if the cloud radar itself moves vertically: the vertical velocity superimposes the measured Doppler velocity. In the case of a moving RV, the vertical movement is induced by the RVs heave rate and rotation. The necessary heave correction was done in a post-processing procedure which will now be introduced.

To enable the correction, the complete cloud radar Doppler spectra as well as the motion data (rotation and translation) of Polarstern were stored with a resolution of 4 Hz and 20 Hz, respectively, throughout the entire cruise. The heave rate of the cloud radar $v_{C_z}$ was determined by summing up the z-component of the cross product between the rotation vector of Polarstern $\boldsymbol{v_{P_R}}$ and the position of the radar $\boldsymbol{X_R}$ relative to the mass centre of the RV

$$\boldsymbol{v_R} = \boldsymbol{v_{P_R}} \times \boldsymbol{X_R} = \begin{pmatrix} P_{pitch} \\ P_{roll} \\ P_{yaw} \end{pmatrix} \times \begin{pmatrix} x_R \\ y_R \\ z_R \end{pmatrix} = \begin{pmatrix} v_{R_x} \\ v_{R_y} \\ v_{R_z} \end{pmatrix} \tag{1}$$

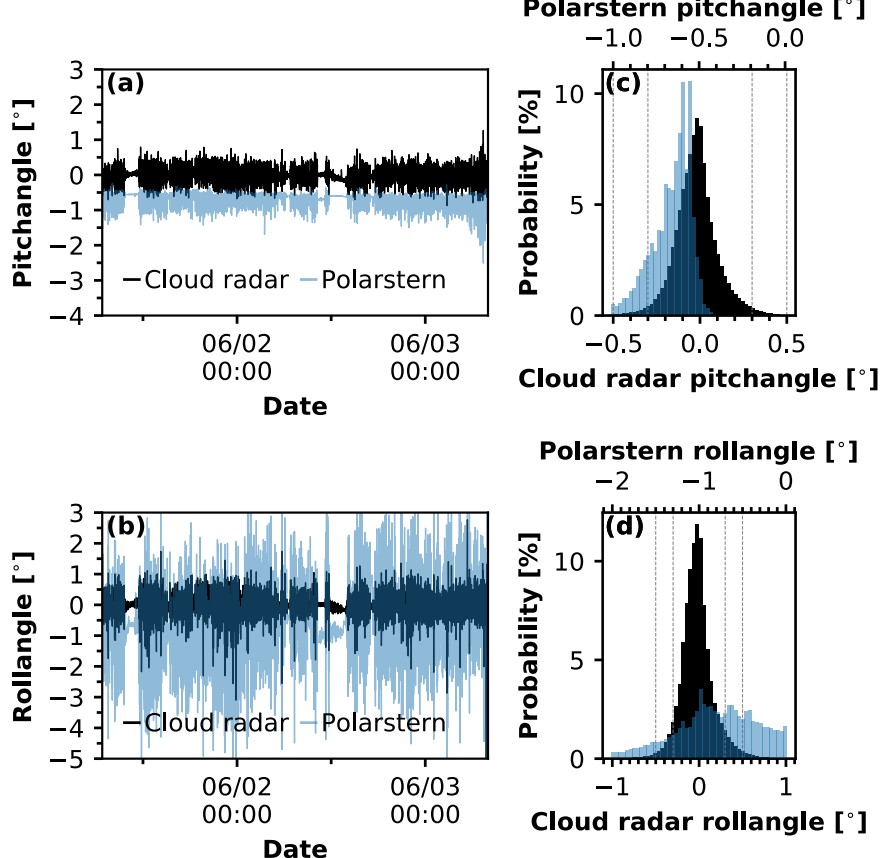

**Figure 4.** Pitch (a) and roll (c) of Polarstern (blue) and cloud radar (black) during the ice transit of Polarstern from 1 June 2017 07:00 UTC – 3 June 2017 8:00 UTC. In (b) and (d) the respective histogram is shown (note the different axes scale of the cloud radar data (bottom axis of each histogram) and Polarstern data (top axis)). The dashed lines indicate a rotation angle of $\pm 0.5°$ and $\pm 0.3°$.

with the z-component of the translation vector of Polarstern $v_{P_{T,z}}$

$$v_{C_z} = v_{R_z} + v_{P_{T,z}}. \tag{2}$$

In Figure 3 (b) the time series of $v_{C_z}$ for PS106 is shown. The heave rate correction was done by shifting each individual Doppler spectrum opposite to the cloud radar heave rate. An illustration of this procedure is shown in Fig 5. In an initial step, the cross correlation between the timestamps of the two data sets, the cloud radar Doppler spectrum and the cloud radar heave rate, was calculated to check for a possible time shift between both data sets. This was found to be 0.25 s. Subsequently, the two values of $v_{C_z}$ from before and after the current Doppler spectrum have been linearly interpolated onto its respec-

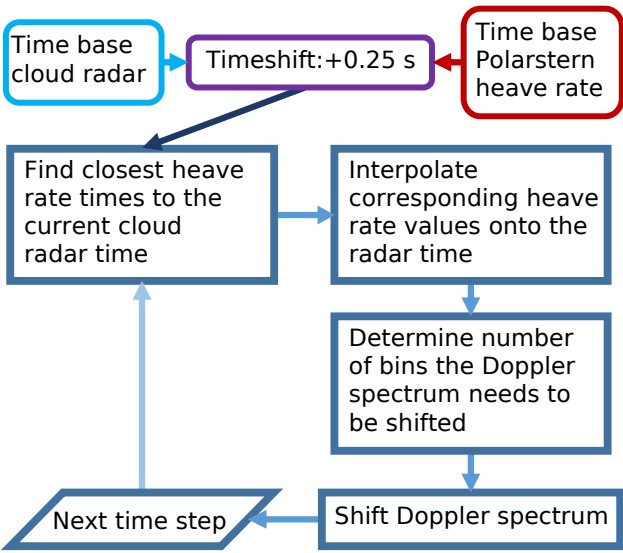

**Figure 5.** Flowchart of the heave rate correction.

tive time. Finally, the spectrum was shifted according to the number of spectral bins determined by the Doppler resolution (

$\Delta v_{DopplerSpectrum} = 0.08 \, \mathrm{m \, s^{-1}}$) and the interpolated heave rate.

The effect of the heave correction is illustrated in Fig. 6. In Figure 6 (a) the uncorrected Doppler velocity measured on 30 May 2017 between 00:00 – 01:00 UTC, together with the respective histogram of the velocities is shown. The RV's movement is visible in both, in the time-height cross-section of the Doppler velocity as stripes of enhanced or reduced velocity throughout the whole column as well as in the broadening of the histogram. The same is presented in Fig. 6 (b) but for the

corrected Doppler velocity.

To evaluate the effect of the heave correction, we calculated the Fourier spectrum of the corrected and uncorrected Doppler velocity (Fig. 6 (e+f)). Continuous time series of 1 hour of Doppler velocity in the highest possible range gates of the cloud were analyzed. To quantify the impact of the heave correction the integral of the frequency range which was most affected by the ship's movement (0.1 – 2 Hz) was calculated both for the corrected and for the uncorrected data. The heave correction

reduced the fraction of the ship's movement in the power spectral density of the cloud radar Doppler velocity by a factor of 15.

## 3.2 Retrieval of eddy dissipation rate

The rate at which turbulence kinetic energy is transferred from larger eddies into smaller ones and eventually dissolve into thermal energy is the EDR. This is used as a quantitative proxy of atmospheric turbulence. Several approaches to retrieve the EDR are common. Methods exist for in-situ measurements from aircraft- (Nicholls, 1978; Nucciarone and Young, 1991;

Meischner et al., 2001), helicopter- (Siebert et al., 2006a), and balloon-borne (Caughey et al., 1979; Siebert et al., 2006b), as well as for meteorological tower instruments (Caughey et al., 1979; Kaimal et al., 1976; Zhou et al., 1985). Additional retrievals

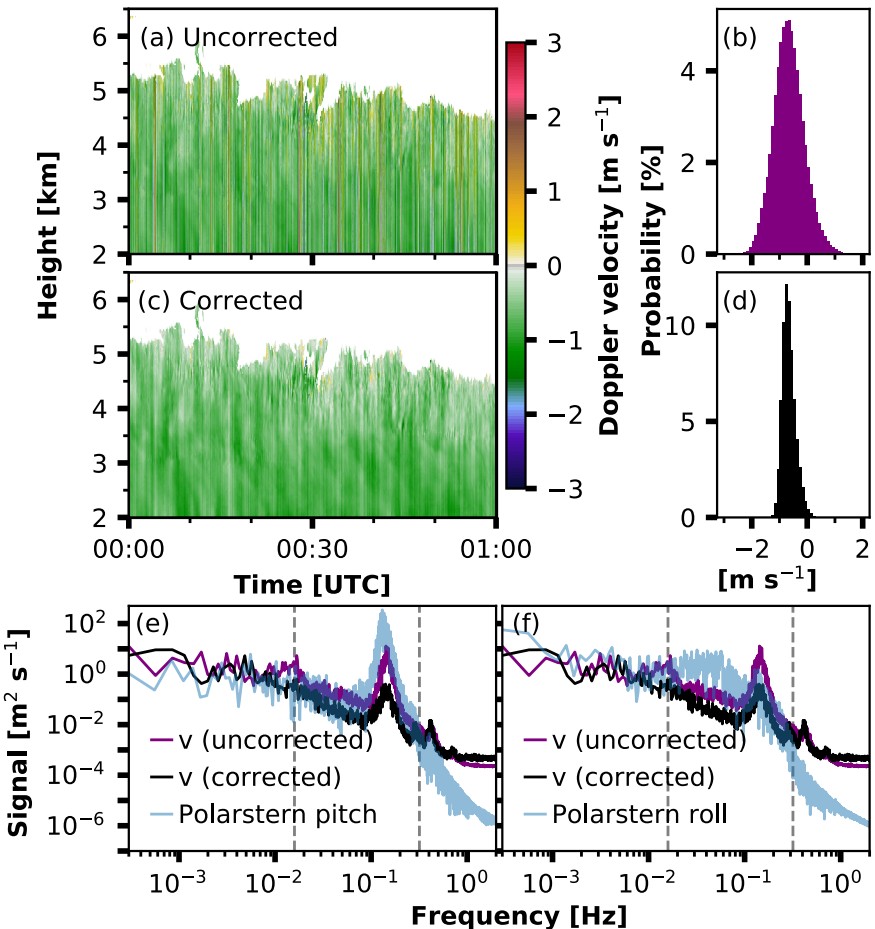

**Figure 6.** Uncorrected (a) and corrected (c) Doppler velocity during PASCAL on 30 May 2017 between 00:00 – 01:00 UTC. (b) and (d) represent the respective histogram of the presented Doppler velocity. Negative values denote downward motion. In (e) and (f) the mean Fourier spectrum of the uppermost, continuous time series of the (uncorrected) Doppler velocity during the same period in black (purple) is shown. In (e) in addition the spectrum of the Polarstern pitch movement during this period is depicted. In (f) the respective spectrum of the roll movement is shown. The dashed lines in (e) and (f) indicate the frequency range which was used to quantify the effect of the heave correction.

for remote sensing observation have been developed (Borque et al., 2016; Sathe and Mann, 2013). These methods are based on the Doppler velocity structure function derived from vertically-pointed Doppler lidar (Frehlich and Cornman, 2002) or Doppler radar (Lothon et al., 2005) or a combination of the width of the Doppler spectrum and the Doppler velocity measurements (Meischner et al., 2001). Other retrievals use time series analyses of vertical velocities from vertical-stare Doppler radar (Shupe et al., 2012; Kalesse and Kollias, 2013) or Doppler lidar observations (O'Connor et al., 2010).

Typical values for EDR in clouds spread between $10^{-1} - 10^{-8}\,\mathrm{m^2\,s^{-3}}$. Borque et al. (2016) report EDR of maritime and continental stratiform clouds in the order of $10^{-4} - 10^{-2}\,\mathrm{m^2\,s^{-3}}$ and $10^{-7} - 10^{-2}\,\mathrm{m^2\,s^{-3}}$, respectively. In cumulus clouds with weak updrafts, EDR had been found in a range between $5 \cdot 10^{-5} - 10^{-2}\,\mathrm{m^2\,s^{-3}}$, whereas values up to $10^{-1}\,\mathrm{m^2\,s^{-3}}$ were found for cumulus clouds with strong updrafts (Siebert et al., 2006a). In cumulonimbus clouds, Meischner et al. (2001) found values for EDR between $10^{-6} - 5 \cdot 10^{-2}\,\mathrm{m^2\,s^{-3}}$. For low clouds or fog at Chilbolton, UK, O'Connor et al. (2010) estimated the EDR to be in the order of $10^{-4} - 5 \cdot 10^{-2}\,\mathrm{m^2\,s^{-3}}$.

The presented range of EDR for different cloud conditions suggests that also Arctic clouds might show characteristic differences for varying atmospheric conditions. The vertical alignment of the cloud radar during PS106 enables the determination of EDR from the vertical air motions observed in cloud layers. Below, we thus present a retrieval technique for EDR that can be applied to the OCEANET data set.

### 3.2.1 EDR from vertical-stare Doppler velocity power spectra

Assuming the turbulent energy dissipation is a homogeneous and isotropic process, the turbulent energy spectrum $S(k)$ within its inertial subrange is represented according to Borque et al. (2016) by

$$S(k) = A\varepsilon^{2/3}k^{-5/3}, \tag{3}$$

with $A = 0.5$ the Kolmogorov constant for a 1-D wind spectra (Sreenivasan, 1995). $k$ represents the wavenumber, which is related to a length scale $L$ ($k = 2\pi/L$) as well as to frequency $f$ with $k = f/V_h$ and $V_h$ as the horizontal wind speed and assuming a linear wind field. If in a log-log plot the observed spectra within the inertial subrange follows a $-5/3$ slope, $\varepsilon$ can be estimated by

$$\varepsilon = \left(\frac{10^{k_0}}{A}\right)^{3/2} \tag{4}$$

where $k_0$ is the corresponding intercept of the linearized fit.

For this study, power spectra of the Doppler velocity with $4\,\mathrm{Hz}$ of continuous time series covering 5 minutes were calculated. To get the best estimate of the respective inertial subrange, the fit was determined by calculating a linear least-squares regression of the spectrum in 34 different wavenumber intervals. The corresponding wavenumber intervals $\Delta k_i$ are depicted in Fig. 7 (a) together with the spectrum of the vertical velocity observed on 7 June 2017, from 10:28 - 10:43 UTC. Following Borque et al. (2016), a good fit was defined with a slope from the linear regression of -5/3 $\pm$ 20% (-5/3 $\pm$ 1/3). If this criteria was matched within more than one wavenumber interval the mean of all $\varepsilon_i$ for one spectrum was calculated. In order to evaluate the EDR estimated by cloud radar measurements, it was compared to EDR derived from the tethered balloon (Egerer et al., 2019). The time periods used for deriving EDR from the tethered balloon was 15 minutes, during which it was located at a constant height above ground. As a measure to quantify the uncertainties of the two retrievals the standard deviation $\sigma$ of all good fits was calculated.

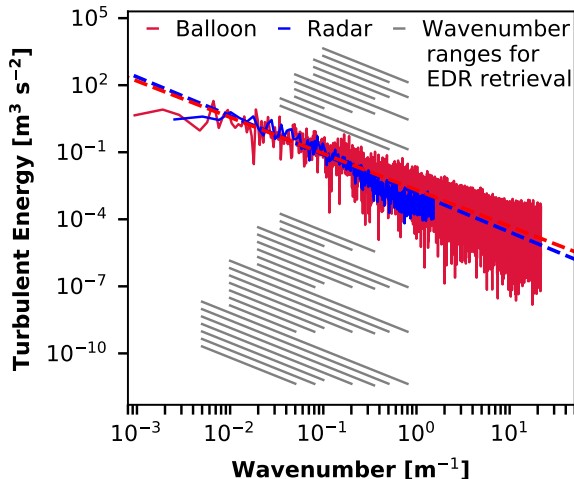

**Figure 7.** Fourier spectrum derived from cloud radar Doppler velocities (blue) and from tethered balloon (red) turbulence measurements on 7 June 2017 between 1027–1043 UTC at 380 m height with their respective averaged linearized fit depicted by the dashed lines. The EDR values of the two methods were: $\varepsilon_{TetheredBalloon} = 2.65 \cdot 10^{-4}\,\mathrm{m^2 s^{-3}}$ and $\varepsilon_{CloudRadar} = 6.84 \cdot 10^{-4}\,\mathrm{m^2 s^{-3}}$ with standard deviation $\sigma_{TetheredBalloon} = 3.59 \cdot 10^{-5}\,\mathrm{m^2 s^{-3}}$ and $\sigma_{CloudRadar} = 7.61 \cdot 10^{-4}\,\mathrm{m^2 s^{-3}}$. Grey lines: Illustration of the wavenumber intervals that had been used to check for a -5/3 slope of the Fourier spectrum.

Three comparisons had been done for situations where the tethered balloon was parked at a constant height within a cloud. One on 7 June 2017, with the tethered balloon being at 380 m height between 10:28 and 10:43 UTC. In Figure 7 (a) an inter-comparison of the power spectrum derived by the tethered balloon measurements (red) with the spectrum derived from the cloud radar Doppler velocity (blue) according to the techniques described above is shown for this period. The other two comparisons were done on 5 June 2017, between 13:50 and 14:05 UTC at 330 m height and on 9 June 2017 09:00 – 09:15 UTC at 500 m height. The EDR values for these cases from the tethered balloon measurements were $8.90 \cdot 10^{-5} \pm 1.07 \cdot 10^{-5}\,\mathrm{m^2 s^{-3}}$ and $6.39 \cdot 10^{-5} \pm 5.48 \cdot 10^{-6}\,\mathrm{m^2 s^{-3}}$ while the cloud radar measurements gave $5.98 \cdot 10^{-5} \pm 3.53 \cdot 10^{-5}\,\mathrm{m^2 s^{-3}}$ and $2.26 \cdot 10^{-5} \pm 1.64 \cdot 10^{-5}\,\mathrm{m^2 s^{-3}}$, respectively.

The comparisons of the two retrievals showed that the values differ by a factor of 2–3. This discrepancy is in the order of magnitude as one could expect due to the spatial distance between the two measurements alone (about 200 m). The dashed line in Fig. 7 (a) is an example for the -5/3 slope and the black lines indicate the wavenumber intervals that had been used to check for a -5/3 slope of the Fourier spectrum.

### 3.3 Cloud macro- and microphysical properties from instrument Synergies

To acquire a data set suitable for the statistical evaluation of the macro- and microphysical properties of clouds observed during PS106, the instrument synergistic approach Cloudnet (Illingworth et al., 2007) was applied. This data set in addition serves to realize model evaluations (Illingworth et al., 2007) and radiative transfer calculations, e.g, with the Rapid Radiative Transfer

Model for climate and weather models (RRTMG; Mlawer et al. (1997); Barker et al. (2003); Clough et al. (2005)). RRTMG is currently utilized for single column radiative transfer calculations. The model considers vertical profiles of relative humidity and temperature, standard atmospheric constituent profiles based on Anderson et al. (1986) and cloud macro and microphysical properties of clouds. These assignments include sets of effective radius and mass concentration of liquid and ice hydrometeors. In the following, the approaches for achieving these data set requirements based on the PS106 remote-sensing observations are described.

### 3.3.1 Cloudnet

The instrument synergy approach Cloudnet (Illingworth et al., 2007) which combines the observations from lidar, cloud radar, microwave radiometer, disdrometer and radiosondes was used to determine cloud physical properties during PS106. To illustrate this procedure, the Cloudnet approach will now be briefly introduced. The measurements are first averaged on a common grid with a vertical and temporal resolution of 31.18 m (resulting from the cloud radar resolution) and 30 s. To estimate the temperature at the respective time-height pixel, radiosonde-based profiles of thermodynamic variables are interpolated on the Cloudnet grid. If no radiosonde was launched from the RV but Polarstern was in the vicinity of Svalbard, soundings from Ny Ålesund (Maturilli, 2017) were substitutionally utilized. As a last fall-back option, data from the Global Data Assimilation System model (GDAS) with a horizontal and vertical resolution of 1° and 3 h (GDAS1) was used as meteorological input into Cloudnet.

Based on the observations scaled on the Cloudnet grid a categorization bit mask is derived, which assigns a series of 7 distinct features to the observed targets: clear yes/no; liquid yes/no; falling yes/no; wet bulb temperature below 0 °C yes/no; melting layer yes/no; aerosol yes/no; insects yes/no. The bitwise categorization ensures that each data point is characterized by a defined combination of these features. The detailed definition of the respective categorization bits is beyond the scope of this paper and has already been given by Hogan and O'Connor (2004).

Based on the individual combination of the categorization bits, an atmospheric target classification is derived as follows: 'Clear sky' is defined as no bit is set for the respective pixel. 'Cloud droplets only' are identified by only the droplet bit being set. The falling bit alone identifies 'Drizzle or rain'. Droplet and falling bit together are interpreted as 'Drizzle/rain & cloud droplets', falling and cold bit together as 'Ice'. Droplet, falling and cold bit combined give 'Ice & supercooled droplets'. The melting bit being set alone identifies 'Melting Ice', and together with the droplet bit the pixel is defined as 'Melting ice & cloud droplets'. The aerosol and insects bit then are accordingly interpreted as 'Aerosol', 'Insects' or 'Aerosol & insects'. Following previous studies (e.g. Shupe, 2011; Mioche et al., 2015) we have defined mixed-phase clouds when (supercooled) liquid water and ice particles are detected in the same data point and when an ice cloud was observed with a liquid or mixed-phase cloud top layer.

Besides the phase of the cloud, the respective mass concentrations of ice and liquid water are determined were applicable. The liquid water content (LWC) is derived by scaling the MWR liquid water path adiabatically onto the cloud pixels defined as liquid or mixed-phase (Frisch et al., 1998; Merk et al., 2016). For pure-liquid data points, the approach of Frisch et al. (2002) is used to derive the cloud droplet effective radius from the observed radar reflectivity factor and liquid water path and

an assumed width of the log-normal cloud droplet size distribution (which was, according to Miles et al. (2000), set to 0.35 in our study). The ice water content (IWC) is calculated using an empirical formula from Hogan et al. (2006) relating cloud radar reflectivity $Z$ and temperature $T$. This approach for IWC is only applied for clouds Cloudnet classified as 'Ice' or 'Ice & supercooled droplets'. In this step also a correction for potential attenuation of the cloud radar signal due to the presence of liquid water is made.

### 3.3.2 Ice crystal effective radius

As discussed above, Cloudnet offers a variety of retrievals for ice microphysical parameters. Nevertheless, the continuous application of radiative transfer calculations requires a consistent availability of ice and liquid hydrometeor effective radius and mass concentration. While Cloudnet already contains retrievals for effective radius and mass concentration of liquid droplets, as well for ice water content, so far no operational retrieval for ice effective radius is available. We therefore decided for the implementation of a new approach which is based on the combination of a definition of the effective radius as the ratio of the third to the second moment of the particle size distribution (PSD) and an empirical relationship between the visible extinction coefficient $\alpha$, cloud radar reflectivity $Z$, and model temperature $T$. Similar as for IWC (and $\alpha$), $r_{e_{ice}}$ is only calculated for datapoints where Cloudnet classified 'Ice' or 'Ice & supercooled droplets'.

Using the ratio of the second to the third moment of the PSD, the effective radius $r_{e_{ice}}$ can be related to IWC and $\alpha$ (Delanoë et al., 2007). This yields for $r_{e_{ice}}$:

$$r_{e_{ice}} = \frac{3}{2} \frac{IWC}{\rho_i \alpha} \cdot 10^6 \ (\mu\text{m}), \tag{5}$$

with $\rho_i$ as density of the solid ice ($\rho_i = 917\,\text{kg m}^{-3}$). Both, IWC and $\alpha$ have been calculated using empirical relationships between IWC or $\alpha$, and the cloud radar reflectivity $Z$ of a 35-GHz cloud radar and temperature $T$ (Hogan et al., 2006).

Finally, we found for the ice crystal effective radius a $Z - T$ relationship:

$$r_{e_{ice}} = \frac{3}{2\rho_i} 10^{C_{ZT} \cdot ZT + C_Z \cdot Z + C_T \cdot T + C} \cdot 10^6 \ (\mu\text{m}), \tag{6}$$

with $C_{ZT} = -2.05 \cdot 10^{-4}$, $C_Z = 1.6 \cdot 10^{-3}$, $C_T = -1.71 \cdot 10^{-2}$ and $C = -1.52$.

To estimate the error of the identified effective radii of the ice crystals, an error propagation of Eq. (6) had be done using the respective error for IWC and $\alpha$ from Hogan et al. (2006).

### 3.3.3 Detection of low-level stratus clouds

During PS106, frequently low-level stratus clouds (cloud base $< 165$ m) have been observed. These situations were often associated with a strong attenuation of the lidar beam within the lowest few hundred meters above Polarstern due to the high optical thickness of these clouds. The cloud radar, in turn, has its technical limitation in detecting the lowest part of the boundary layer below 155 m range (165 m above sea level). Due to the instrument synergy approach of Cloudnet this is also the height of the

lowest Cloudnet data pixel. Thus, the low-level clouds which occurred during PS106 introduced on the one hand issues to the Cloudnet retrieval due to misinterpretation of attenuated lidar signal as missing signal. On the other hand, since most current statistics of Arctic clouds do not consider clouds in such a low altitude, these clouds tend to be underrepresented in Arctic cloud statistics. To address these issues, we introduce a new Cloudnet classification category called low-level stratus cloud. These

335 clouds were identified by the Polly$^{XT}$ signal-to-noise ratio (SNR, (Heese et al., 2010)) in the lowest 165 m above sea level. The near-range channels of the Polly$^{XT}$ system have a complete overlap already at 120 m above the instrument (Engelmann et al., 2016) and thus are suitable for the detection of clouds already well below the lowest cloud radar observation height, even though quantitative parameters such as (attenuated) backscatter coefficient from a single elastic backscatter signal cannot yet be determined at these heights.

From visual inspection of the Cloudnet data set we defined low-level stratus where the SNR exceeded the threshold value of 40. This value was obtained by evaluating signatures of attenuation in the time series of the Cloudnet attenuated backscatter coefficient, increased values of LWP time series, and correlation with the visibility sensor of Polarstern. Since the SNR is not yet range-corrected, this threshold-crossing at these low altitudes is very likely only due to the occurrence of low-level clouds. The low level stratus base and top have been derived by simply using the lowest and highest range gate from Polly$^{XT}$ where

the SNR exceeded the threshold.

### 3.4   Retrieval of CCN- and INP number concentrations

Arctic clouds and their susceptibility to the presence of aerosol are still in focus of research (Morrison et al., 2012). Based on the measurements of Polly$^{XT}$, an estimation of cloud condensation nuclei (CCN) and ice nucleating particle (INP) properties is possible (Mamouri and Ansmann, 2016). To do so, profiles of the aerosol backscatter coefficient and depolarization ratio

need to be determined. In a second step, these profiles are converted into profiles of the particle extinction coefficient using an appropriate lidar ratio (extinction-to-backscatter ratio).

    The CCN number concentration (CCNC) and INP number concentration (INPC) profiles were estimated from profiles of the lidar-derived particle extinction coefficient at 532 nm by means of conversion parameters and published INP parameterization schemes (DeMott et al., 2010) as described by Mamouri and Ansmann (2016). The required conversion parameters for Arctic

AERONET stations were determined in the same way as outlined by Mamouri and Ansmann (2016). We used multi-year (2004-2017) Sun-photometer observations of the AERONET stations Thule, PEARL, Kangerlussuaq, Ittoqqortoormiit, and Hornsund to obtain the set of Arctic conversion parameters. These AERONET observations were made during the summer half years.

    The direct retrieval of the CCN conversion parameters from the AERONET data (level 2, version 3, inversion products)

revealed $C_1 = 18.6\,\mathrm{cm}^{-3}$ and exponent $d_1 = 0.83$ for the range of extinction coefficients from $15 - 300\,\mathrm{Mm}^{-1}$ (500 nm AOD from $0.015 - 0.3$ were measured). During the PS106 observations, the aerosol extinction coefficient was mostly around $1-10\,\mathrm{Mm}^{-1}$ in the lower part of the troposphere. The AERONET data for this low range of extinction coefficients indicates that conversion parameters of $C_2 = 10\,\mathrm{cm}^{-3}$, $d_2 = 0.9$, $C_3 = 3.0\,\mathrm{cm}^{-3}$, $d_3 = 1$ would be appropriate. The aerosol in the Arctic

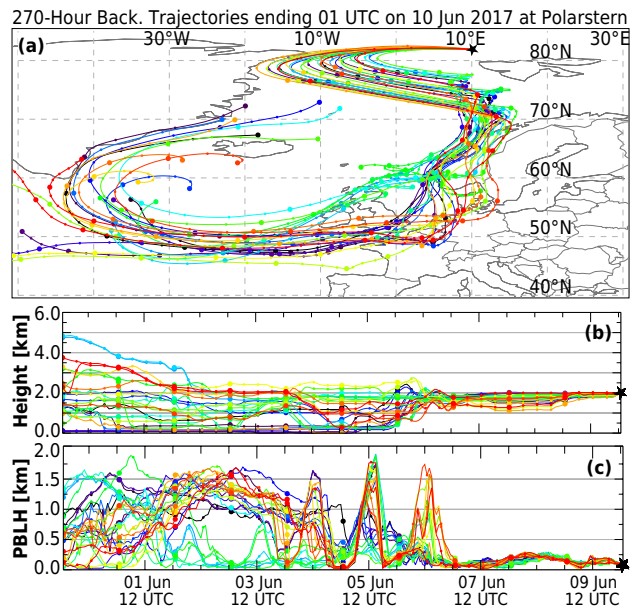

**Figure 8.** (a): Ensemble of 27 10-day back-trajectories arriving at the position of Polarstern at 01 UTC on 10 June 2017 in 2000 m height. (b) shows the height of the trajectory and (c) the respective planetary boundary layer height (PBLH).

is fine-mode dominated and shows Ångstrom exponents (440–870 nm) typically between 0.9 and 1.8 (with an average of 365 1.5–1.6).

## 4 Results

### 4.1 Case studies

Based on near-surface and radiosondes measurements, model data and satellite observations Knudsen et al. (2018) gave a detailed description of the synoptic situation during the PASCAL campaign. They defined three periods (cold, warm and 370 normal period). Three case studies will be presented in the following which are all within the warm period (WP 30 May – 12 June 2017). These two weeks of the WP are characterized by warm and moist air advection from the south crossing Norway and Greenland (Fig. 8).

The three cases between 7 June and 9 June 2017 were chosen to demonstrate the potential OCEANET offers. Within these three days the near-surface temperature first dropped from -3.5°C on 7 June 2017 00:00 UTC to -7.5 °C on 8 June 2017 375 05:00 UTC with an ensuing increase to 1.0°C on 9 June 2017 22:00 UTC due to warm air advection. These cases were, on the one hand, selected to represent the capabilities of the standard Cloudnet products and of Polly$^{XT}$ and on the other hand to illustrate the new products introduced in this study.

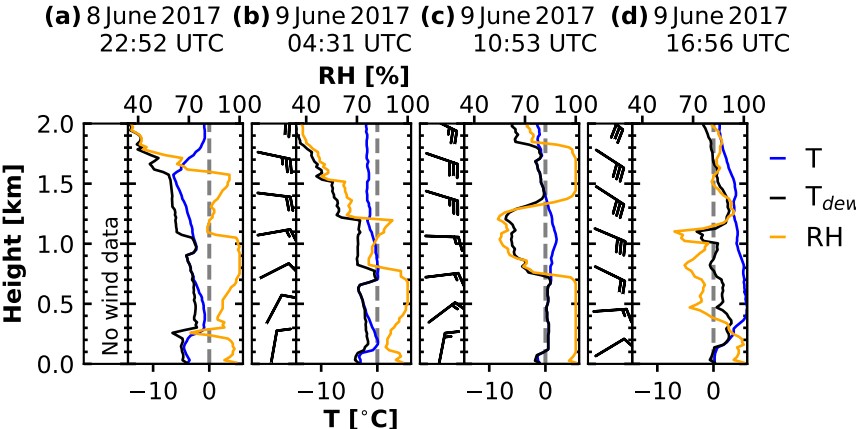

**Figure 9.** Thermodynamic profiles from radiosondes launched aboard Polarstern on (a) 8 June 2017 at 22:52 UTC, (b) 9 June 2017, 04:31 UTC, (c) 9 June 2017, 10:53 UTC, and (d) 9 June 2017,16:56 UTC up to 2 km height. Each sounding is divided into two parts: left side the wind barbs, right side the temperature (blue), dew point temperature (black), and relative humidity (orange) profiles.

### 4.1.1 Precipitating layered cloud: 9 June 2017, 00:00 – 18:00 UTC, ice floe camp

An overview of the capability of the OCEANET measurements and its application to analyze cloud and aerosol structures and their interactions will be presented for a complex case that occurred over Polarstern on 9 June 2017 between 00:00 – 18:00 UTC. The radiosonde profiles for this period are shown in Fig. 9 up to a height of 2500 m. The observations of the cloud radar, the lidar and the MWR are depicted in Fig. 10.

The presented day reveals a rather complex situation. Starting at 00:00 UTC, Cloudnet classified a liquid stratocumulus layer between 600 – 900 m height with a cloud top temperature of -1.5 °C. This layer slowly descended, reaching a cloud base of about 400 m and cloud top of about 800 m at 05:00 UTC. The LWP during this period was rather constant with a mean value of 50 g m$^{-2}$ with two distinct peaks: one at around 01:50 UTC and the other one around 03:45 UTC with a LWP of up to 70 g m$^{-2}$, both associated with a slight increase in cloud depth. The constantly high values of EDR until roughly 05:00 UTC ($10^{-4} - 10^{-3}$ m$^2$s$^{-3}$, Fig. 11 (d)) indicate strong turbulent mixing of the cloud layer.

At around 05:30 UTC, a transformation of the cloud occurred. The LWP increased up to 160 g m$^{-2}$ and precipitation started, almost reaching the ground (the disdrometer aboard Polarstern showed no precipitation signal, not shown here). Though the LDR showed no increased values, the presence of ice was identified due to detection of enhanced radar reflectivity and vertical velocity and thus a mixed-phase cloud was classified between 05:30 – 06:30 UTC. During this period, the retrievals of the ice and liquid hydrometeor size as proposed in this paper may be influenced by each other. Both retrievals are based on the same quantity, the radar reflectivity, which is characterized by the largest peak in the Doppler spectrum. To tackle this issue, a peak separation of the Doppler spectrum as it is proposed e.g., by Shupe et al. (2004), Kalesse et al. (2016b) or Radenz et al. (2019) would be necessary. This would offer the opportunity to calculate the effective radius of the different hydrometeors species

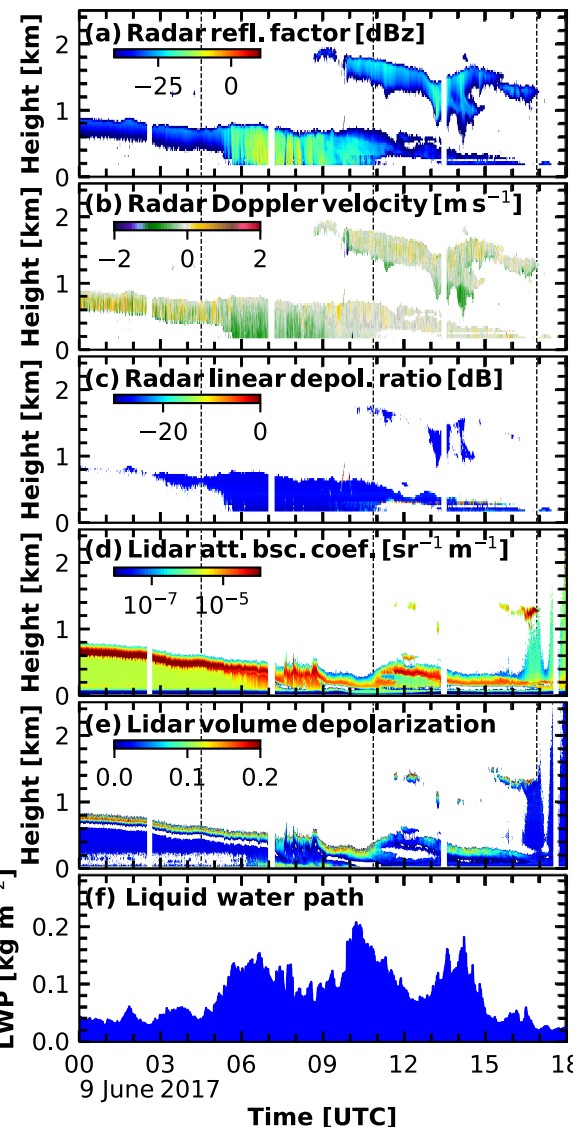

**Figure 10.** OCEANET observations on 9 June 2017 between 00:00 – 18:00 UTC. (a), (b) and (c) show the radar reflectivity factor, Doppler velocity, and linear depolarization ratio. (d) and (e) depict the lidar attenuated backscatter coefficient at 1064 nm and volume depolarization ratio at 532 nm. In (f) the liquid water path derived by the microwave radiometer is shown. The dashed vertical lines mark the time of the radiosonde launches on 9 June 2017 (note: the time of the first launch shown in Fig. 9 was before the plotted profiles of the measurements start).

based on their particular reflectivity but is beyond the scope of this paper. At around 06:30 UTC the interpolated temperature

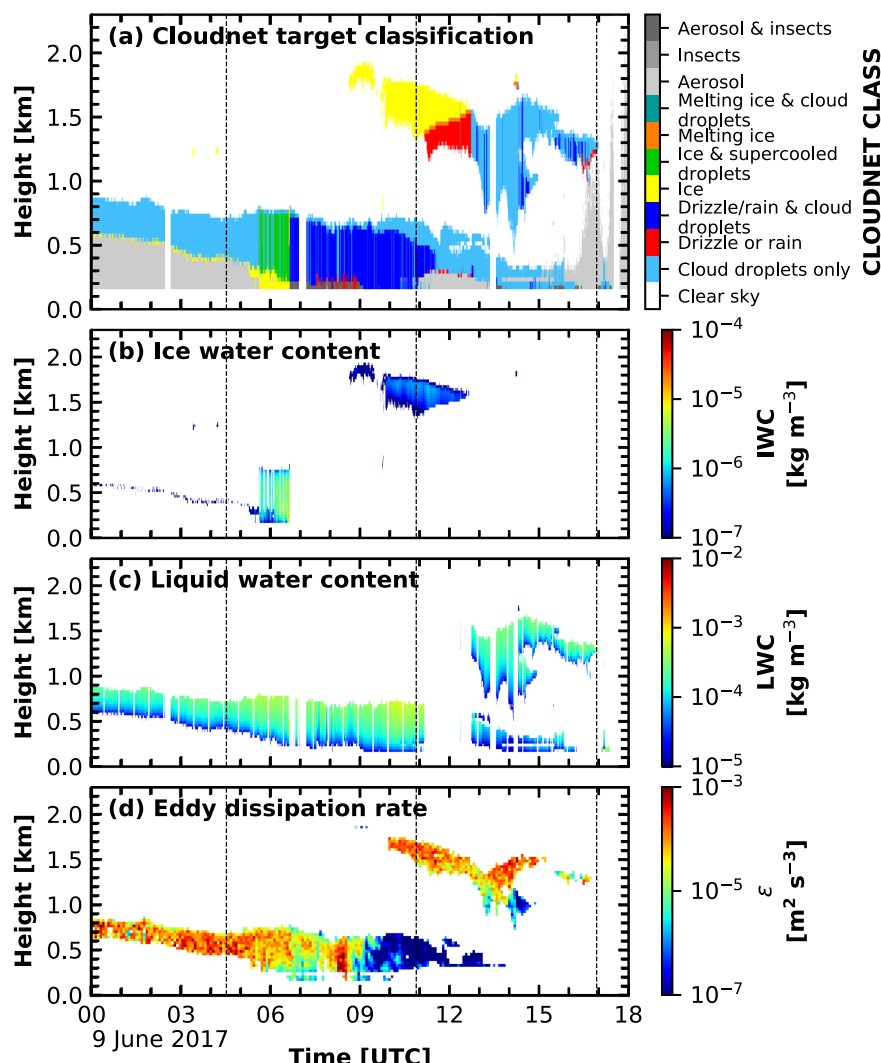

**Figure 11.** (a) Cloudnet target classification, (b) ice water content, and (c) liquid water content for 9 June 2017 between 00:00 – 18:00 UTC. (d) shows the time-height profile of EDR calculated from the cloud radar Doppler velocity. The dashed lines mark the time of the radiosonde launches as shown in Fig. 9.

of the surrounding radiosonde profiles reached 0 °C leading to an immediate transition from a mixed-phase to a pure liquid cloud. Therefore, no IWC and no ice effective radius were determined under these conditions.

A second transition of the cloud situation during this day is associated with an altocumulus layer which was located above the stratocumulus. Around 09:00 UTC this mid-level cloud layer with a cloud-top temperature of -1°C occurred at 1900 m height over Polarstern. As this layer increased in geometrical and optical depth, shading effects reduced the cloud-top radiative cooling of the cloud below as already observed in (Shupe et al., 2013). This led to a collapse of the EDR in the lower layer at around 12:00 UTC (Fig. 11 (d)) and finally to a dissipation of the cloud. The values for $\varepsilon$ in the altocumulus were about the

same order of magnitude as for the stratocumulus, indicating that the upper cloud was able to effectively cool to space. Starting at about 14:00 UTC the altocumulus formed a two-layer structure at 1500 m and 1200 m, respectively. Due to the shading of the upper layer, the lower one lost its turbulent moment and the cloud dissipated shortly after. The altocumulus was classified as pure ice cloud, probably due to the fact that the lidar signal was already fully attenuated in the lower layer which impedes the classification as liquid at an ambient temperature below 0°C. At around 11:00 UTC, the temperature exceeded 0°C, due to

which Cloudnet changed its classification from an ice cloud to a liquid cloud. After persisting for about 4 hours with rather low EDR, the stratocumulus started to dissipate at around 16:00 UTC. This offered the lidar the opportunity to observe the aerosol structure above Polarstern in the subsequent hours (Fig. 12 (b)).

### 4.1.2 Aerosol case: 9 June 18:00 UTC – 10 June 2017 11:00 UTC, ice floe camp

Between 9 June 18:00 UTC – 10 June 2017 11:00 UTC, one of the rare cloud-free events of PS106 occurred and Polly[XT]

observed aerosol layers in the free troposphere (Fig. 12 (d)). The respective radiosonde profiles for this period are shown in Fig. 12 (a–c). Based on a trajectory analysis of 27 10-day HYSPLIT back-trajectories (Stein et al., 2015), a southern inflow for air masses above the boundary layer is identified for this period. At the 2000 m height level, the trajectories show that these were long-range-transported aerosol layers that originated over continental Europe (Fig. 8) with a high chance of being within the planetary boundary layer at that time. Below 2000 m height, the trajectories indicate that pathways mainly crossed the north

sea and the Atlantic ocean (not shown).

The 1064-nm lidar attenuated backscatter coefficient and the 532-nm volume depolarization ratio are shown in Fig. 12 (d, e). These measurements reveal the existence of three aerosol layers being present over Polarstern on 9 June 2017 around 18:00 UTC. A shallow layer at 500 m height staying roughly at the same altitude as long it was observed by Polly[XT]. A second one is visible between 1000 and 1500 m height ascending to 2500 m at 00:00 UTC on 10 June 2017. At 07:00 UTC on 10 June

2017, a liquid cloud formed at the top of this layer. A third aerosol layer with a liquid cloud embedded at 2300 m height, being rather constant in altitude, was present between 19:00 – 21:00 UTC on 9 June 2017.

In Figure 13, a detailed analysis of the aerosol optical properties as derived from the lidar measurements from the time period of 00:00 – 02:20 UTC is presented. During this period, two layers were detected and are visible in the profiles of the backscatter coefficient at all three wavelengths (Fig. 13 (a)). The rather strong wavelength dependence of the backscatter

coefficient, as shown by the high Ångstrom exponents (Fig. 13 (b)) in both layers, indicates the presence of small aerosol particles. A back-trajectory analysis and the values for the Ångstrom exponent indicates that air masses transporting polluted

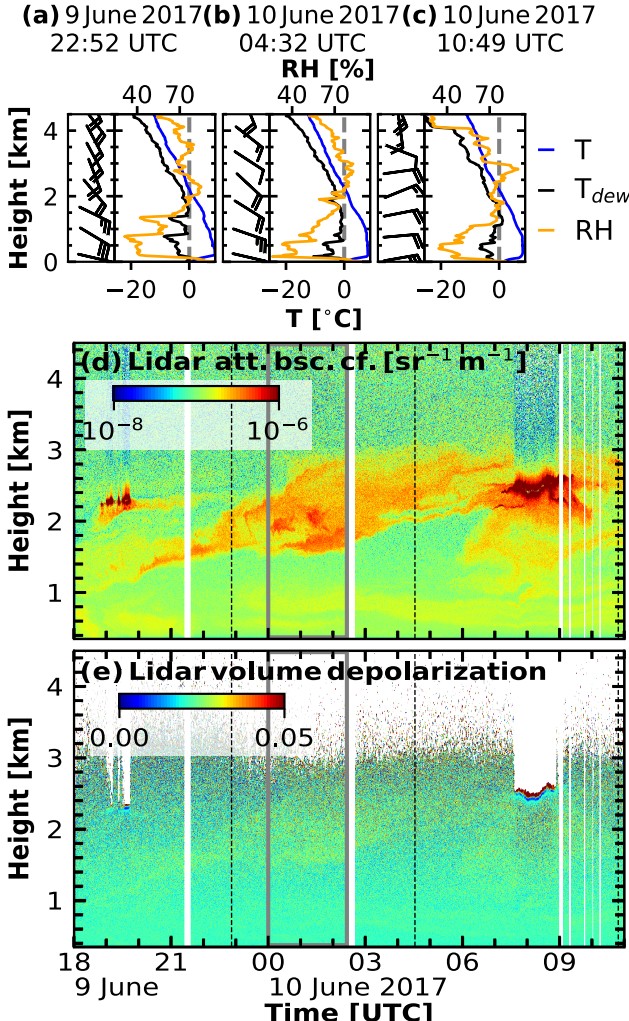

**Figure 12.** (a-c) same as Fig. 9 but for 9 June 2017 18:00 UTC – 10 June 2017 11:00 UTC up to 5 km height. (d+e) measurements from Polly$^{\text{XT}}$ between 9 June 2017, 18:00 UTC – 10 June 2017, 11:00 UTC. In (d) and (e), the 1064-nm attenuated backscatter coefficient and the 532-nm volume depolarization are shown, respectively. The black dashed lines mark the time of the radiosonde launches as shown in (a–c). The grey box indicates the period used for averaging in Fig. 13.

aerosol from continental Europe are most probably the source for the upper aerosol layer. The lower aerosol layer on the other hand is most-likely a mixture of down-mixed continental and upward-mixed marine aerosol.

Based on the aerosol optical properties retrieved by Polly$^{\text{XT}}$ an estimation of the CCNC was done for all three combinations of conversion factor and extinction exponent as mentioned in Sect. 3.4. We chose the second combination ($C_2 = 10\,\text{cm}^{-3}$ and $d_2 = 0.9$) to illustrate the results in Fig. 13 (d). The mean values of CCNC for the upper aerosol layer in this case was found to be $\bar{n}_{CCN,2} = 180\,\text{cm}^{-3}$. For the two other combinations, we found $\bar{n}_{CCN,1} = 260\,\text{cm}^{-3}$ and $\bar{n}_{CCN,3} = 75\,\text{cm}^{-3}$,


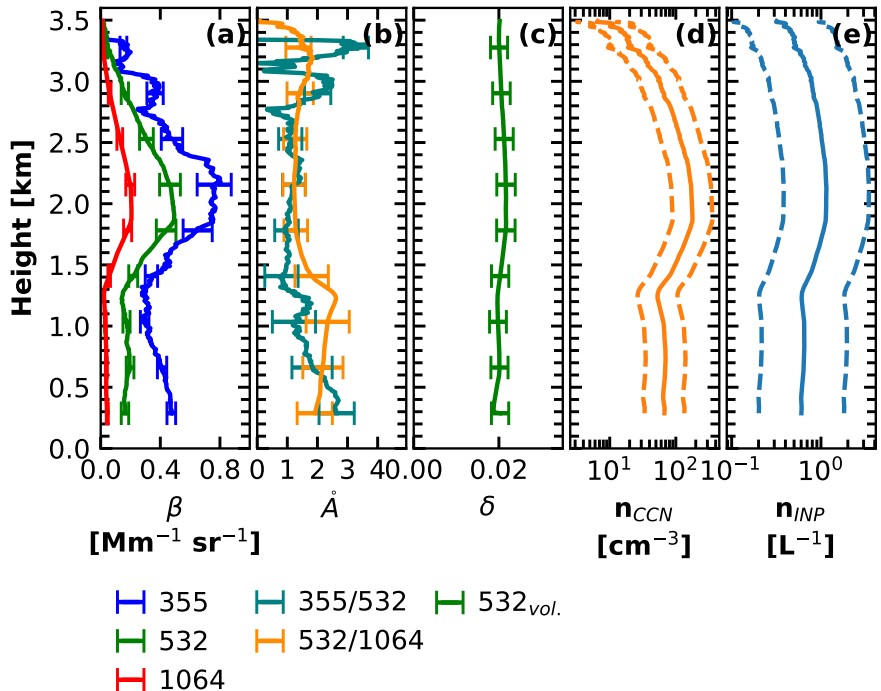

**Figure 13.** Averaged aerosol optical properties for the time period from 00:00 – 02:20 UTC on 10 June 2017 up to 3.5 km height. (a) aerosol backscatter coefficient for three wavelengths of 355 nm (blue), 532 nm (green) and 1064 nm (red). (b) Backscatter-related Ångstrom exponent for 355 nm to 532 nm (cyan) and for 532 nm to 1064 nm (orange). (c) 532-nm volume depolarization ratio. In (d) the retrieved CCN number concentration for $C_2 = 10\,cm^{-3}$ and $d_2 = 0.9$ and in (e) the INP number concentration for $T = -15°C$ is shown together with the respective uncertainty (dashed lines) derived from the 532-nm backscatter coefficient profile shown in (a).

respectively. For the lower aerosol layer, we found $\bar{n}_{CCN,2} = 70\,cm^{-3}$. For the two other combinations, we found $\bar{n}_{CCN,1} = 110\,cm^{-3}$ and $\bar{n}_{CCN,3} = 25\,cm^{-3}$, respectively. The uncertainty of this method is up to 200% (dashed lines in Fig. 13 (d)). In addition, the INPC was calculated for a fixed temperature of -15°C. The results are shown in Fig. 13 (e). The INPC for the lower layer was found to be around $0.6\,L^{-1}$ for this temperature. In the upper layer, $n_{INP}$ went up to values slightly above $1\,L^{-1}$. These calculations have an error of a factor of three (dashed lines in Fig. 13 (e)) but nevertheless provide a guideline about the conditions of the cloud-relevant aerosol properties on the discussed day.

### 4.1.3 Ice cloud: 7 June 21:00 UTC – 8 June 09:00 UTC, ice floe camp

In Figure 14, we present the OCEANET measurements for the period from 7 June, 21:00 UTC to 8 June 2017, 09:00 UTC. The corresponding thermodynamic profiles from the radiosondes launched during this time period are shown in Fig. 15. In Figure 16 (a), the Cloudnet target classification, together with the IWC (b), ice effective radius (c) and ice water path (d) are

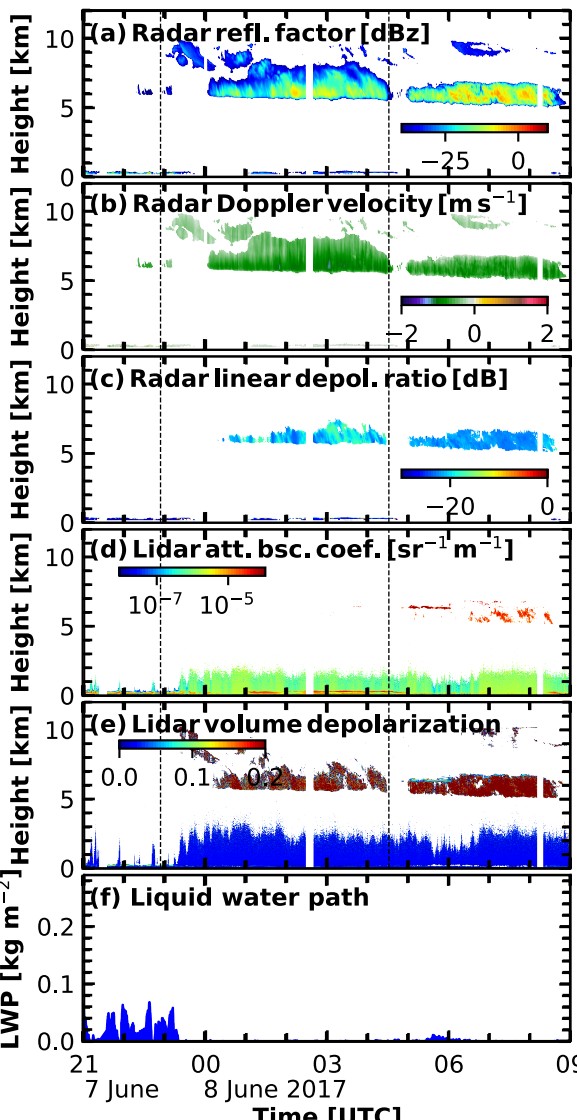

**Figure 14.** Same as Fig. 10 but for 7 June 2017 21:00 UTC − 8 June 2017 09:00 UTC (note: the first and last launch shown in Fig. 15 was before the plotted profiles of the measurements start).

shown. The ice water path is derived as the integral of the IWC for each profile. This period is chosen to illustrate the low-level stratus cloud detection algorithm and the retrieval of the ice effective radius.

At the beginning of the addressed time period, a low-level mixed-phase stratiform cloud reaching up to a height of 500 m is present. This layer is visible in both the cloud radar reflectivity as well as in the lidar attenuated backscatter data. Due to its high optical thickness, this cloud led almost continuously until 23:30 UTC to an attenuation of the lidar signal already close to

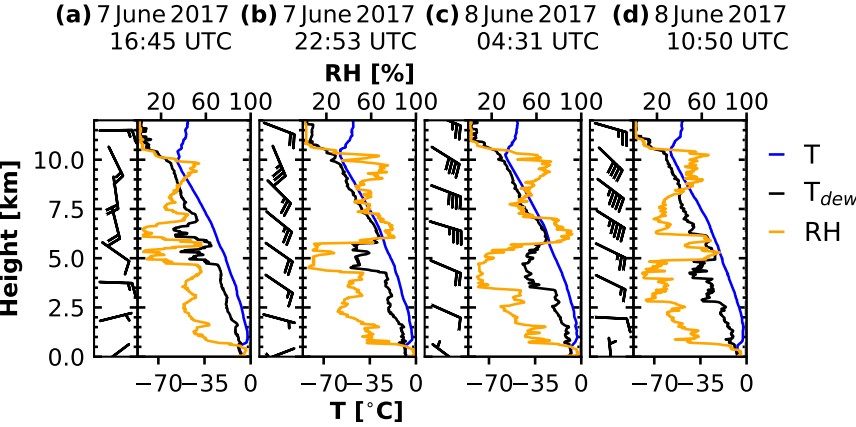

**Figure 15.** Same as Fig. 9 but for 7 June 2017 21:00 UTC – 8 June 2017 09:00 UTC up to 12 km height.

cloud base (Fig. 14). Only occasionally, backscattered lidar signals from aerosols above the cloud were detected. During this period, the liquid water path varied between values of around $0\,\mathrm{g\,m^{-2}}$ for moments when the lidar was able to detect signal from above the cloud and values up to $100\,\mathrm{g\,m^{-2}}$ associated with periods when the lidar signal was attenuated already close to cloud base.

In Figure 17, the derived low-level stratus classification mask (below 165 m height) combined with a simplified Cloudnet target classification mask (above 165 m height) for this period is shown. Red areas depict detected low level clouds. Blue and green data points indicate clear sky and aerosols, respectively. Though Polly[XT] detected low-level stratus almost continuously during the case study, this affected the lidar signal most severely during the above mentioned period. After 23:30 UTC, the LWP showed values of around $10\,\mathrm{g\,m^{-2}}$ and the cloud lost most of its optical thickness so that the lidar was able to penetrate through the cloud.

Well above the low-level stratus some cirrus clouds formed around 22:30 UTC above 6000 m height. These transformed into a cirrostratus layer at 00:00 UTC which was present between 6000 and 10000 m height. Until around 04:30 UTC, this cloud is classified as a pure ice cloud, characterized by LDR values of up to -15 dB and a constantly downward directed vertical velocity, with a tiny patch of detected liquid at around 03:30 UTC at 6100 m height.

At 04:30 UTC, the cirrostratus dissipated and another layer started to pass over Polarstern at around 05:00 UTC. This layer with coexisting liquid droplets and ice crystals extended from 5000 m up to 7000 m height. While the cloud radar reflectivity factor was higher in this layer compared to the first one, the cloud radar LDR decreased to values of below -20 dB. On top of this layer a supercooled liquid layer was detected by the lidar between 05:00 – 06:00 UTC, characterized by high attenuated backscatter coefficient and low values of linear depolarization ratio. Additionally, some regions with high linear depolarization ratio were detected by Polly[XT] inside the cirrus after 06:30 UTC, probably associated to a mixture of supercooled droplets and ice crystals.

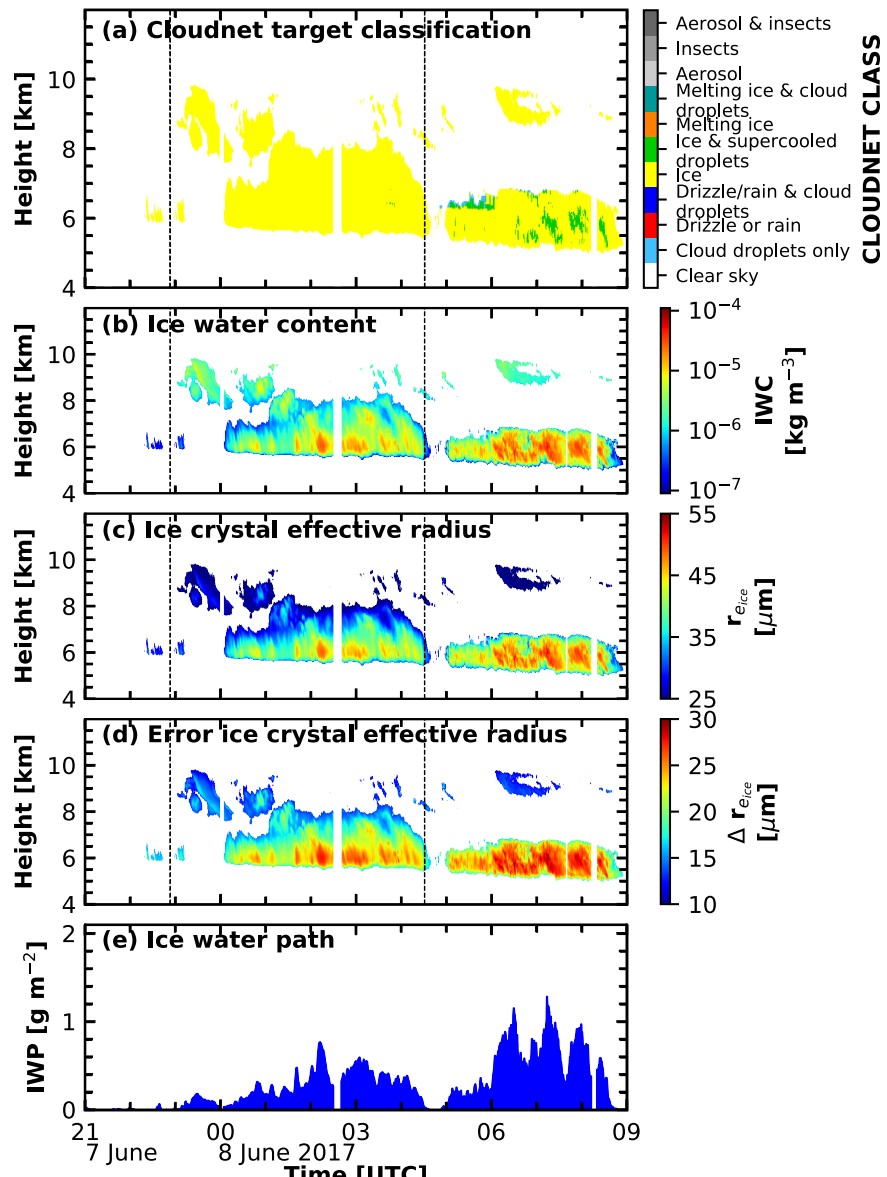

**Figure 16.** Cloudnet products for 7 June 2017, 21:00 UTC to 8 June 2017, 09:00 UTC: (a) target classification, (b) ice water content, (c) ice crystal effective radius, (d) uncertainty of the retrieved ice crystal effective radius as derived from error propagation, and (e) ice water path. The dashed lines mark the times of the radiosonde launches shown in Fig. 15 (note: the time of the first and last launch shown in Fig. 15 was before and after the presented time period).

The IWC of the cirrostratus was in a range from $10^{-4} - 10^{-6}\,\mathrm{kg\,m^{-3}}$ with lowest values at cloud top and highest values at cloud base. The ice effective radius ranged from $30 - 55\,\mu$m and its distribution follows the same pattern as the one of the IWC, as can be expected because both follow a similar reflectivity-temperature relationship.

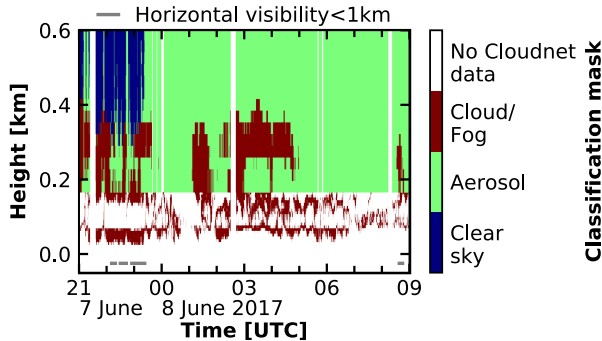

**Figure 17.** Low-level cloud mask for 7 June 2017 21:00 UTC – 8 June 2017 09:00 UTC derived from combining Polly[XT] and Cloudnet data. Below 165 m height, red colors indicate when low-level stratus was detected using the signal-to-noise ratio (SNR) of the Polly[XT] 532 nm near-field channel. Above 165 m height, a simplified version of the Cloudnet target classification mask is shown. Everything which was detected as cloud (either ice or liquid or mixed-phase) is masked in red. Blue depicts clear sky and green aerosols. The light-blue line at the bottom indicates periods when fog was detected by means of the visibility sensor aboard Polarstern.

## 4.2 Cloud statistics

In Figure 18, an overview about the statistical distribution of the cloud occurrence during PS106 is given. In Figure 18 (a), daily statistics of the vertical distribution of low-level stratus is shown. In addition, the frequency of occurrence of this cloud type for each day is illustrated in Fig. 18 (b). Low-level stratus was detected during a significant period of time on almost each day. The highest frequency of occurrence was observed while the Polarstern was surrounded by sea ice. Rather low values occurred while Polarstern was in the vicinity of Svalbard. In order to asses our retrieval of the low level clouds we plotted in comparison the frequency of occurrence of vertical visibility below 1 km.

Statistics of the cloud type occurrence are shown in Fig. 18 (c). The daily frequency of occurrence as well as the total distribution for the complete campaign of low level stratus clouds (purple), liquid clouds (orange), ice clouds (light blue), mixed-phase clouds (green), multi-layer clouds (dark blue), and cloud-free situations (yellow) is shown. In addition, an analysis of the co-occurrence of low-level stratus and other cloud types was performed and is shown in the very right column of Fig. 18 (c). The rate of coexistence of the respective cloud type together with low-level stratus is indicated by a slightly varied color code.

In total, during 11% of the time cloud-free conditions were detected by Cloudnet during PS106. The two most prominent cloud types were multi-layer and mixed-phase clouds with an occurrence frequency of 38.5% and 36% of the observational time, respectively. Pure ice clouds were present for about 8% and pure liquid clouds for about 4.5% of the time, respectively. Single events of the new Cloudnet class low-level stratus cloud were detected during 2.5% of the time of the two month campaign. In addition, 27% of the observed liquid clouds and 48% of the ice clouds occurred simultaneously with low-level

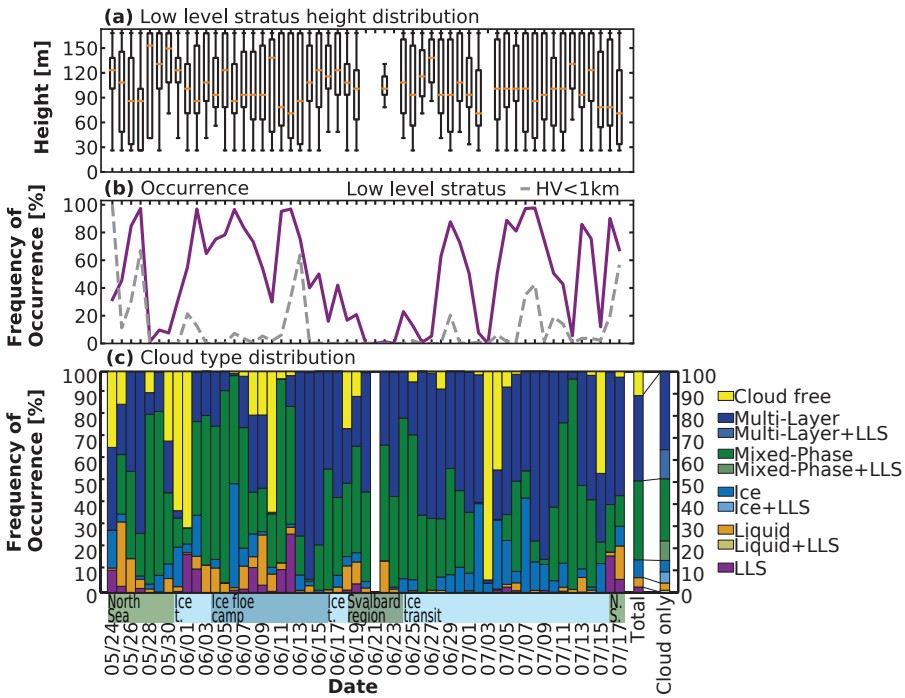

**Figure 18.** (a) daily height distribution of the detected low level stratus during PS106 up to 165 m. (b) daily fraction of low level stratus occurrence derived by Polly$^{XT}$ measurements (purple) in comparison to horizontal visibility (HV) below 1 km (grey) . (c) cloud type statistics including low level stratus clouds during PS106, determined by Cloudnet. Purple indicates the fraction when low-level stratus was determined, orange liquid clouds, light blue ice clouds, green mixed-phase clouds, dark blue multi-layer clouds and yellow cloud free periods. Each column except the last two represents one day of the campaign. The penultimate column represents the total distribution of the different cloud types. The last column distinguishes between the respective cloud type without low-level stratus detected (same color as in the other column) and with an additional low-level stratus detected below (slightly varied color). At the bottom a rough localization of Polarstern is annotated (green: North Sea (N.S.), light blue: Ice transit (Ice t.), dark blue: Ice floe camp, dark green: Svalbard region).

stratus. Mixed-phase and multi-layer clouds were detected together with low level stratus clouds during 24% and 27% of their respective observational time.

In contrast to Nomokonova et al. (2019), who provided a statistical analysis of the cloud occurrence over Ny Ålesund, Svalbard, for the period between June 2016 and July 2017, we found a higher frequency of single layer mixed-phase clouds at the expense of cloud-free and single-layer liquid clouds when comparing the period of PS106. This may be due to a difference

in turbulence as well as in a change of the cloud microphysics at locations surrounded by sea ice or open ocean (Young et al., 2016).

## 5 Summary and Conclusions

A two-month campaign of RV Polarstern, including an extensive suite of ground-based remote sensing instruments of the OCEANET platform, has been conducted north- and northeast of Svalbard in the Arctic summer of 2017. This study described in detail the deployed instrumentation and the applied processing schemes. Only few campaigns with a comparable equipment have been performed in recent years at these latitudes, e.g., ASCOS which took place from 2 August to 9 September 2008 (Tjernström et al., 2014) and ASCE in the Arctic summer and early autumn of 2014 (Tjernström et al., 2015). A new feature of PS106 was the deployment of a motion-stabilized vertically-pointing 35-GHz cloud radar during and the correction of the Doppler velocity subsequent to the cruise as specified in Sect. 3.1.

For an automatic, seamless analysis of cloud properties from the measured remote-sensing time series, the Cloudnet algorithm was utilized. In doing so, new products were developed and applied to the remote sensing data set from PS106. This was done in order to enable the continuous characterization of cloud turbulence by means of EDR, and to provide mass concentration and effective radius of ice crystals and liquid water droplets as future input for radiative transfer simulations. Though being well established, applying the Cloudnet algorithm to data from a remote-sensing supersite aboard a research vessel in the Arctic reveals new challenges. The movement of the ship has a significant effect on the measured vertical velocity of the cloud radar. To tackle this issue, the cloud radar was mounted on a stabilization platform to guarantee its vertical pointing. The vertical velocity data set was corrected for the heave rate of the ship in a post-processing procedure subsequent to the cruise.

The motion stabilization was evaluated by means of a small single board computer mounted on the cloud radar rack. The IMU of the mini computer measured the residual of the pitch and roll movement after stabilization. We found a good stabilization during ice breaking conditions with a leveling precision of $\pm 0.5°$. During rough sea however the displacement from zenith was larger, up to $\pm 1°$. Under the strong wave conditions during these time periods it needs also to be considered, that the IMU of the orientation sensor used for the cloud radar is based on so called MEMS (Micro-Electro-Mechanical Systems). Such devices are based on spring-mounted capacitor plates and thus measured pitch and roll angles are affected by translational motions like engine vibrations etc. As these effects were not investigated in the frame of our study, we conclude that the actual vertical-pointing uncertainty range, especially on the open sea, was lower than the one reported by the MEMS sensors, i.e., better than $\pm 1°$.

Using the corrected vertical velocities from the cloud radar, the eddy dissipation rates were calculated and evaluated against in-cloud turbulence measurements done by means of a tethered balloon. This intercomparison revealed a good agreement between both approaches where the values where within the estimated uncertainty and the expected difference due to the spacial distance. Nevertheless the tethered balloon approach seems to be the more reliable one, given the smaller standard deviation.

Based on published retrievals of visible extinction coefficient and ice water content, a new approach to derive the effective radius of the ice crystals was introduced. The associated uncertainties, estimated by error propagation, of the ice crystal effective radii are presented in Fig. 16 (d). On average the uncertainty is about 50% of the size of the radii themselves which reflects the strong influence of uncertainties in the underlying observational data on the retrieval. Given the challenges in estimating the

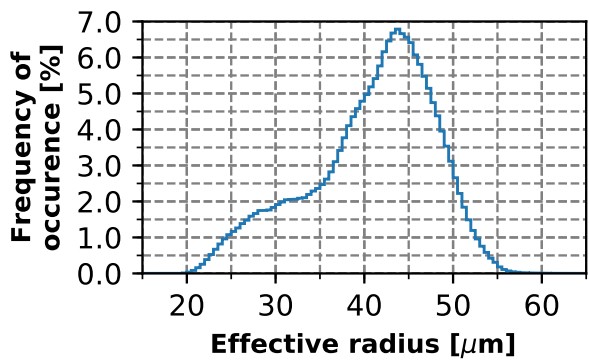

**Figure 19.** Histogram of the ice particle effective radius for PS106. Integration over x yields 100%.

effective radius of ice crystals on a continuous basis on the one hand and the necessity of having such values, e.g., for radiative transfer calculations, on the other hand, we consider this estimate to be still in a reasonable range. In Figure 19, the histogram of the effective radius for full PS106 is shown. Values range from $20 - 60\,\mu$m, with a peak at around $50\,\mu$m. This is consistent with other studies of ice effective radius (e.g., Blanchard et al., 2017).

This study revealed in addition the relevance of the lowest detection limit of remote sensing instruments on the representativity of Arctic-cloud statistics. Cloudnet is configured to have its lowest range gate at the lowest detection altitude of the cloud radar, which was 165 m above the ocean surface for PS106. Lower-level cloud layers are thus not identifiable within Cloudnet. In our study, lower cloud structures were identified using the SNR measured by the lidar Polly[XT]. This ability has been used to study the occurrence of low-level stratus clouds below the first Cloudnet range gate. Sotiropoulou et al. (2014) used a
combination of cloud radar and ceilometer measurements to study stratiform Arctic clouds and found that most stable, surface coupled clouds have a cloud base below 200 m. Yet, so far such clouds have not been considered in many Arctic cloud climatologies derived by remote sensing instruments. Liu et al. (2012) for example defined low-level clouds as those between 0 and 2000 m, with 960 m above the ground being the height where surface contamination effects on Cloudsat become insignificant, and using a vertical resolution of 240 m. Shupe (2011) summarized cloud statistics from several multi-year data sets derived
from ground-based remote-sensing observations for different sites in the Arctic. He specified a height dependence of cloud occurrence down to 300 m by using a combination of lidar and radar. Below 300 m, however, he provided information about cloud occurrence but without any further specification of the cloud base. Even airborne remote sensing instruments suffer from the strong ground clutter and thus struggle to deliver information about cloud occurrence below 150 m height above the surface (Mech et al., 2019). The autonomous buoys of the IAOOS network equipped with a microlidar observed in 2014 and 2015 in
the Arctic clouds with a base height below 500 m during 60 % of their observational time Mariage et al. (2017).

Our study shows that a higher vertical resolution and reliable signal from very low altitudes is required to characterize the lowest-level cloud layer which occur between approximately 50 m and 165 m above ground. Such clouds stay undetected for ground-based in-situ sensors (because they are too high) as well as for most automatized ground-based remote sensing

instruments (because they are too low). Future radiative transfer studies should show what the effect of the lowest-level clouds,

which occurred during 25% of the observation time, is on the radiation budget of the region where PS106 was performed.

Future work will confront the observed cloud macro- and microphysical properties as well as the EDR with high resolution model simulations along the PS106 track that have been carried out in the framework of (AC)[3]. The herein introduced remote sensing techniques will also be applied to the data set of the currently ongoing one-year polar ice drift of RV Polarstern during the MOSAiC project (Schiermeier, 2019), thus providing an unprecedented data set of Arctic aerosol and mixed-phase clouds.

This data set will substantially contribute to our understanding of the role of clouds in the current warming of the Arctic climate system.

*Code and data availability.* The radiosonde data is available by Schmithüsen (2017a) (PS106.1) and Schmithüsen (2017b) (PS106.2). The lidar measurements are available by Griesche et al. (2019), the cloud radar measurements by Griesche et al. (2020b). The Cloudnet data set is available by Griesche et al. (2020a) and related data sets. The publication of the low level stratus cloud data set is in progress Griesche and

Seifert (2020).

*Author contributions.* PS, RE and JB designed the OCEANET measurements and PS, RE, JB, MR and HG prepared the instruments. HG, RE, MR and CB operated the OCEANET instruments aboard Polarstern. PS and HG processed the data with Cloudnet and developed the low level stratus detection algorithm. HG performed the heave correction and developed the EDR and ice crystal effective radius retrieval with supervision from PS, RE and JB. HB, ZY, and AA provided lidar-based data processing for retrieval of aerosol optical and microphysical

properties. AM provided support in the data analysis and manuscript preparation. HG drafted the manuscript with contributions from all co-authors.

*Competing interests.* The authors declare that they have no conflict of interest.

*Acknowledgements.* We gratefully acknowledge the funding by the Deutsche Forschungsgemeinschaft (DFG, German Research Foundation) – Project Number 268020496 – TRR 172, within the Transregional Collaborative Research Center "ArctiC Amplification: Climate Relevant

Atmospheric and SurfaCe Processes, and Feedback Mechanisms (AC)3". We thank the Alfred Wegener Institute and R/V Polarstern crew and captain for their support (AWI_PS106_00). We also thank the University on Cologne for providing us with the MWR retrieval for Ny-Ålesund, Svalbard, Norway. Authors acknowledge also support through ACTRIS-2 under grant agreement no. 654109 from the European Union's Horizon 2020 research and innovation programme.

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
