# Peer review of "Application of the shipborne remote sensing supersite OCEANET for profiling of Arctic aerosols and clouds during Polarstern cruise PS106"

_Atmospheric Measurement Techniques, 2019_

## Referee Comment (RC1) · Anonymous Referee #1 · 16 Jan 2020

General comments This paper presents the instruments deployed on the icebreaker Polarstern and close by on a temporary ice-camp and results obtained during a summer cruise performed in the frame of the AC3 German project in 2017. Several remote sensing equipment, including a motion-stabilized 35-GHz cloud radar were deployed and combined with meteorological observations in high Arctic. This experiment concurred to a very important goal on a better documentation and understanding of Arctic change, through the presentation of a campaign and results obtained to better document Arctic cloud forcing. After introducing the context, this paper first gives a general description of the instrumentation deployed on board the ice-breaker Polarstern and on the ice camp, technical challenges, new developments, analysis methods and results

obtained during the campaign. It finally focuses on case studies. Two main points are highlighted in the paper which are 1) the first involvement of the cloud radar Mira-35 and the development of a motion stabilization system to ensure stable observations. Corrections and results obtained from vertical wind spectra to derive on the turbulent kinetic energy eddy dissipation rate (EDR) are presented; 2) the focus on low-level clouds and the presence of fog from synergies of lidar and radar within Cloudnet, and the retrieval of radiative cloud properties.

The topic is of importance to the community. The paper is clearly written, and presented in a very comprehensive way. The context of the paper is well introduced although additional general information should be given on existing surface based observations. The two main points presented also need some additional information and discussion. The paper is worth publishing after minor revisions are made. They are addressed here below.

Detailed comments Page 2, line 27 : "decline of the Arctic sea ice" precision to be added on period of the year (summer ?) or ice type (multi-year sea-ice) ?

Page 2, line 45-47 : Arctic observations refer to aircraft and shipborne measurements, but Arctic ground-based stations should be discussed ( IASOA network, Uttal et aL, BAMS 2016 DOI:10.1175/BAMS-D-14-00145.1) in which remote sensing instruments are implemented at Barrow (Dong et al., 2010, doi:10.1029/2009JD013489, Eureka (Blanchard et al., JAMC 2014 doi: 10.1175/JAMC-D-14-0021.1) for example. Drifting buoys have also recently been equipped in the high Arctic with lidar in the frame of the IAOOS project (DiBiagio et al., JGR 2018, doi: 10.1002/2017JD027530 ; Mariage et al., Opt. Exp. 2016, doi: 10.1364/OE.25.000A73).

Page 2, line 54, replace by a more recent reference Winker et al., BAMS, 2010, doi:10.1175/2010BAMS3009.1.

Page 4, line 88 : Figure 1 legend : mark days also on the track in the upper figure

Page 4, line 94 : mention if Polar measurements have already been performed ?

Page 5, lines 103 and 104 : 532 instead of 512 ?

Page 5, line 110: "allow to determine the shape" this is too strong a statement. As the authors write further in the text, it allows to discriminate shape between spherical and non-spherical particles, but several shapes can give the same depolarization ratio

Page 6, lines 125-26 : the authors "do think that the atmospheric conditions in summer in the Arctic are comparable to those in winter in the Netherlands ". I don't think so. Surface temperature are close to zero over ice and surface-atmosphere interactions are different

Page 7, Table 1 : Add information on the auxiliary measurements (tethered balloon, sonic anemometer, pyranometers, ...)

Page 8, Figure 3 legend : extend period limits on the vertical with dotted lines

Page 10, Figure 5 : put the histograms outside the figure so to better see the full 2D plot

Page 10, line 194 to 246 : Extend discussion on error induced by the correction. What is the expected in terms of residual contribution ? What bias is to be considered in the sigma correction, and error induced as an additional error. This can be discussed from the spectrum shape, errors and confidence in the limits of analysis to be used. Present/discuss more in detail the corrected spectrum in section versus non-corrected one and versus the sonic anemometer one.

Page 11, line 230 : typo vertical

Page 12, Figure 6 shows linearized fit from sonic only, what would be the one from corrected spectrum ? Discuss values retrieved from the range of the fit identified from the residual errors and confidence in the correction.

Page 12, Figure 6 legend : refers to values of EDR, but hypotheses for deriving EDR

from radar should be more discussed (see above).

Page 12, line 251 : Iacono et al., 2208, is not a general reference for RRTMG. This ref is to be replaced by a more appropriate one.

Page 13, line 288 : It is OK here, but more generally for Arctic louds I am not sure of that, as for supercooled precipitating clouds

Page 14, line 310 : a strong attenuation

Page 14, line 318 : I would suggest to use scattering ratio Sr as well, which would further allow to discuss fog issue using lidar measurements only assuming a threshold in Sr

Page 15, line 348 : I would suggest to extend presentation here and discuss meteorological context change to introduce cases studies and overall meteorological patterns observed leading to the various cases analyzed. I would suggest to move Figure 11 here and briefly discuss more general transport evolution over the period studied (not necessarily adding a figure).

Page 20, Figure 10 : I would suggest to present lidar scattering ratio instead of backscattering coefficient (to better support aerosol/fog/cloud discrimination).

Page 23, and 24 : Synergies between the remote sensing instruments and auxiliary observations from aboard Polarstern were analyzed by means of Cloudnet classification procedure. This procedure is shown to induce caveats because of limitations in the radar range measurements. More discussions on the way this could be mitigated using lidar measurements should be included.

Page 23, lines 429-432 : PollyXT "Though detected fog almost continuously during the case study, . . .". How is this done ? Explain in the text how this can be translated in an additional information below 165 m in a quantitative way from scattering ratio.

Page 24, Figure 15 : Blue color below 165 m shows occurrence of clear air <165m. It is

thus misleading as no information is available from Cloudnet. Should be another color corresponding to unknown (white?) instead of blue below 165 m in Fig 15. Could be replaced by dots corresponding to fog color on a white background from the discussion on fog detection by lidar only.

Page 24 line 435 : "above the fog layer" meaning well above !

Page 28, lines 509-510 : Yes, frequently observed from surface-based IAOOS observations as reported in Mariage et al., 2016

---

## Referee Comment (RC2) · Anonymous Referee #2 · 21 Jan 2020

The authors describe the deployment of the Oceanet remote-sensing container during a cruise to the Arctic. Right now it is not clear if the authors want to present technical development or research findings. The authors briefly describe a new motion stabilisation platform and a new data processing method for fog detection. However, they fail to provide a validation that those are working. The remainder of the paper is dedicated to case studies. The paper is of interest to the community but needs major revisions. First of all, the authors need to make up their mind if this should be a paper for AMT or ACP. There are further major items that need to be addressed before it can be considered for publication:

[Figure]

- [Abstract] The Abstract appears to be more of an introduction than a concise summary of the paper. Key points of the article are missing. Please rewrite the Abstract.

- [MWR retrieval] One of the major issues with this work is related to the analysis of the microwave radiometer measurements. I do not agree with the assumption that atmospheric conditions in the Arctic are comparable with those during winter in the Netherlands. In the Netherlands the minimum temperature rarely reaches $0°C$; also radiative balance is not comparable. The analysis needs to be repeated with a customised Arctic retrieval. Radiosonde data can be obtained from several research cruises in the Arctic since 1990 and are also available from research stations around the Arctic.

- [Motion stabilisation] The authors should provide proof that the roll and pitch was actively levelled out for the motion-stabilised radar measurement. Please provide a time series of roll angles for the ship and radar during roughest sea and the probability distribution of radar roll angle for at least a 1 h period with greatest ship roll. Further information on the measurement conditions is needed to assess the performance of the motion stabilisation platform. What was the maximum roll angle? What was the ship's mean horizontal velocity when underway? What was the wave-induced velocity perturbation in open water?

- [Eddy dissipation rate] The validation of eddy dissipation rate is not convincing. At what height was the tethered balloon located? Below a cloud or within a cloud? What are the reasons for the over- and underestimation? Also, it would be good to have more than two comparisons cases between the Radar and the measurements with the tethered balloon or to provide justification why this is not done. Please also provide the ÉŻ values from tethered balloon and radar for both cases.

- [Cloudnet and cloud definition] There are several issues related to the Cloudnet retrieval. Right now it is often unclear what has been done. For instance, the description of the classification mask (Page 13, line 261) does not agree with the shown Cloudnet target classification in Figure 9a. Please provide more details on Cloudnet in general

and on the classification mask and the target classification for readers that are not familiar with the method. Further, it is not clear if the presented definition of liquid and mixed-phase clouds (page 12, second paragraph) is an official Cloudnet product such as the target classification or if it is a new data product developed by the authors. In that context, why not use the target classification as in comparable studies based on multi-sensor retrievals? In those, Arctic mixed phase clouds are defined when both liquid/supercooled water and ice particles are present and when ice particles are identified directly below liquid and mixed-phase regions (e.g. Shupe 2011, Mioche et al. 2015). For comparison of cloud statistics from different campaigns it is important to use the same definition as already used in the literature.

- [Fog detection] The information related to the fog detection is not adequate to evaluate if the proposed method works. Please make use of the visibility sensor aboard Polarstern to assess your findings as well as to test if your assumed SNR value of 40 can be used to reliably detect fog. Just as a reminder, fog is defined when the visibility is below 1 km. The visibility sensor can also be used to distinguish between fog and low clouds. In that regard, please compare the detected low cloud layers with the observation of the ceilometer aboard Polarstern. The first height bin is much lower than the first height bin of the Polly system. Also would it not be better to use the ceilometer for detection of fog and low cloud layers? First of all the first cloud layer is lower and the Ceilometer on Polarstern is a CL51 which reports the vertical visibility in case that the lowest height bins are obscured due to precipitation and/or fog?

Minor issues

- Line 88, Please also cite Ehrlich et al. (2019) for ACLOUD

- Line 122: It is not clear if only winter time radiosondes from De Bilt are used in the retrieval. Please clarify. But even better would be to revise the retrieval using actual Arctic measurements.

- Line 126: An Arctic retrieval based on ERA-Interim data should be compared to

a retrieval based on Radiosonde data. Systematic errors in ERA-Interim data (e.g. Wesslén et al., 2014, for temperature bias) can have an influence on the MWR retrieval. Consider using ERA5 instead.

- line 153: Please provide the typical error range of the RS92 measurements

- line 270: Do you mean T or Td (dew point temperature here).

- line 344: Can you please verify if the mixing depth provided by GDAS1 is comparable to the observed mixing depth. It is known that models have problems to provide realistic mixing depth in the Arctic.

- line 354: Do you mean Figure 4.1.1? And the profiles are shown to a height of 2.5 km not 2.0 km.

- line 356: Since ice particles are below the liquid stratocumulus the cloud should be reclassified as mixed-phase cloud (see major comments).

- line 375: As observed by Shupe et al. (2013). Please add citation.

- The figures do not appear in the order they are discussed in the text. Please revise.

- Figure 6b is not necessary and should be omitted.

- Figure 15: Fog and low cloud should have different colours. Again use the visibility sensor to distinguish between fog and low clouds. Add visibility to the plot. Also add the observed backscatter from the CL51 as comparison as an extra plot next to it.

- Figure 17: How is fog height determined? That needs to be discussed in 3.3.3. Add visibility to the plot.

- line 500, e.g. Sotiropoulou et al. (2014) and (2016) considered low clouds from ceilometer/Halo and radar measurements.

- Line 762: Somag, the provided link does not work. Please provide an open link or add the information to the text.

- Figures 7, 10 (upper panel), and 13: Please use same scale for T and RH in all plots.

Ehrlich, A., Wendisch, M., Lüpkes, C., Buschmann, M., Bozem, H., Chechin, D., . . . Zanatta, M. (2019). A comprehensive in situ and remote sensing data set from the Arctic CLoud Observations Using airborne measurements during polar Day (ACLOUD) campaign. Earth System Science Data Discussions, 1–42. https://doi.org/10.5194/essd-2019-96

Wesslén, C., Tjernström, M., Bromwich, D. H., De Boer, G., Ekman, A. M. L., Bai, L. S., & Wang, S. H. (2014). The Arctic summer atmosphere: An evaluation of re-analyses using ASCOS data. Atmospheric Chemistry and Physics, 14(5), 2605–2624. https://doi.org/10.5194/acp-14-2605-2014

Shupe, M. D.: Clouds at Arctic Atmospheric Observatories, Part II: Thermodynamic phase characteristics, J. Appl. Meteor. Clim., 50, 645–661, doi:10.1175/2010JAMC2468.1, 2011.

Mioche, G., Jourdan, O., Ceccaldi, M., and Delanoë, J.: Variability of mixed-phase clouds in the Arctic with a focus on the Svalbard region: a study based on spaceborne active remote sensing, Atmos. Chem. Phys., 15, 2445–2461, https://doi.org/10.5194/acp-15-2445-2015, 2015.

Shupe, M. D., Persson, P. O. G., Brooks, I. M., Tjernström, M.,Sedlar, J., Mauritsen, T., Sjogren, S., and Leck, C.: Cloud and boundary layer interactions over the Arctic sea ice in late sum-mer, Atmos. Chem. Phys., 13, 9379–9399, doi:10.5194/acp-13-9379-2013, 2013.

Sotiropoulou, G., Sedlar, J., Tjernström, M., Shupe, M. D., Brooks, I. M., and Persson, P. O. G.: The thermodynamic structure of summer Arctic stratocumulus and the dynamic coupling to the surface, Atmos. Chem. Phys., 14, 12573–12592, https://doi.org/10.5194/acp-14-12573-2014, 2014.

Sotiropoulou, G., Tjernström, M., Sedlar, J., Achtert, P., Brooks, B. J., Brooks, I. M.,

Wolfe, D. (2016). Atmospheric conditions during the Arctic clouds in summer experiment (ACSE): Contrasting open water and sea ice surfaces during melt and freeze-up seasons. Journal of Climate, 29, 8721–8744

---

## Author Comment (AC1) · 28 Apr 2020

Dear Reviewer, we are currently working on the revision of the paper. What do you mean with ÉZ? Can you provide a reference where this is explained? Best regards, Hannes Griesche

---

## Author Comment (AC2) · 15 Jun 2020

**Authors response - AMT**

**Application of the shipborne remote sensing supersite OCEANET for profiling of Arctic aerosols and clouds during Polarstern cruise PS106 – Griesche et al.**

**Response to Anonymous Referee #1 (16 January 2020)**

We would like to thank the Anonymous Referee #1 for spending time in order to provide us with fruitful comments and suggestions and thus to help us to improve the manuscript. The initial submission has been adapted, and we hope that the manuscript is now acceptable for publication.

Our point-by-point response to the review comments is written here in **bold** font.

**Overall summary of major changes:**

We would like to inform the referee about the following major changes:

- Revision of the abstract due to suggestion of Referee #2
- Reprocessing of the Cloudnet data with an Arctic MWR-retrieval considering the comments of both Referee #1 and #2
- Included a better evaluation of the capabilities of the motion stabilization according to comments by both Referee #1 and #2
- Improved the discussion of the eddy dissipation rate and fog/low-stratus retrievals as well as Cloudnet in general considering the comments by both Referee #1 and #2

**Detailed responses:**

This paper presents the instruments deployed on the icebreaker Polarstern and close by on a temporary ice-camp and results obtained during a summer cruise performed in the frame of the AC3 German project in 2017. Several remote sensing equipment, including a motion-stabilized 35-GHz cloud radar were deployed and combined with meteorological observations in high Arctic. This experiment concurred to a very important goal on a better documentation and understanding of Arctic change, through the presentation of a campaign and results obtained to better document Arctic cloud forcing. After introducing the context, this paper first gives a general description of the instrumentation deployed on board the ice-breaker Polarstern and on the ice camp, technical challenges, new developments, analysis methods and results obtained during the campaign. It finally focuses on case studies.

Two main points are highlighted in the paper which are 1) the first involvement of the cloud radar Mira-35 and the development of a motion stabilization system to ensure stable observations. Corrections and results obtained from vertical wind spectra to derive on the turbulent kinetic energy eddy dissipation rate (EDR) are presented; 2) the focus on low-level clouds and the presence of fog from synergies of lidar and radar within Cloudnet, and the retrieval of radiative cloud properties.

The topic is of importance to the community. The paper is clearly written, and presented in a very comprehensive way. The context of the paper is well introduced although additional general information should be given on existing surface based observations. The two main points presented also need some additional information and discussion. The paper is worth publishing after minor revisions are made. They are addressed here below.

Detailed comments Page 2, line 27 : "decline of the Arctic sea ice" precision to be added on period of the year (summer ?) or ice type (multi-year sea-ice) ?

We added details "This is observed as a change of several parameters such as the drastic decline of the Arctic sea ice during all seasons, but especially in summer, in both extend and thickness".

Page 2, line 45-47 : Arctic observations refer to aircraft and shipborne measurements, but Arctic ground-based stations should be discussed (IASOA network, Uttal et al., BAMS 2016 DOI:10.1175/BAMS-D-14-00145.1) in which remote sensing instruments are implemented at Barrow (Dong et al., 2010, doi:10.1029/2009JD013489, Eureka (Blanchard et al., JAMC 2014 doi: 10.1175/JAMC-D-14-0021.1) for example. Drifting buoys have also recently been equipped in the high Arctic with lidar in the frame of the IAOOS project (DiBiagio et al., JGR 2018, doi: 10.1002/2017JD027530; Mariage et al., Opt. Exp. 2016, doi: 10.1364/OE.25.000A73). We extended the discussion about ground-based stations and buoy observations, as requested.

Page 2, line 54, replace by a more recent reference Winker et al., BAMS, 2010, doi:10.1175/2010BAMS3009.1.

Page 4, line 88 : Figure 1 legend : mark days also on the track in the upper figure **We added some dates for orientation also into the top subfigure of Fig. 1.**

Page 4, line 94 : mention if Polar measurements have already been performed ? **We added information about previous cruises.**

Page 5, lines 103 and 104 : 532 instead of 512 ? Corrected

Page 5, line 110: "allow to determine the shape" this is too strong a statement. As the authors write further in the text, it allows to discriminate shape between spherical and non-spherical particles, but several shapes can give the same depolarization ratio

We rephrased the respective passage in the text and provide references to the available applications of polarization measurements.

Page 6, lines 125-26 : the authors "do think that the atmospheric conditions in summer in the Arctic are comparable to those in winter in the Netherlands". I don't think so. Surface temperature are close to zero over ice and surface-atmosphere interactions are different

Meanwhile, we reprocessed the MWR data with an retrieval that was created by University of Cologne for the location of Ny Alesund (78.9°N, 11.8°E). We mention this in the revised manuscript. The data will also be uploaded as a new version to Pangaea. Same holds for the depending Cloudnet-processed dataset on Pangaea. Fig. 1 shows the correlation of the two datasets. Overall, the correlation is quite linear, especially for the

Fig. 1: Comparison of LWP (a, b) and IWV/PRW (b) derived from (unflagged) MWR observations with retrievals from De Bilt and Ny Alesund.

Page 7, Table 1 : Add information on the auxiliary measurements (tethered balloon, sonic anemometer, pyranometers, ...) **Done**

Page 8, Figure 3 legend : extend period limits on the vertical with dotted lines **Done**

Page 10, Figure 5: put the histograms outside the figure so to better see the full 2D plot **Done**

Page 10, line 194 to 246 : Extend discussion on error induced by the correction. What is the expected in terms of residual contribution ? What bias is to be considered in the sigma correction, and error induced as an additional error. This can be discussed from the spectrum shape, errors and confidence in the limits of analysis to be used. Present/discuss more in detail the corrected spectrum in section versus non-corrected one and versus the sonic anemometer one. We have extended the discussion and used the Fourier analysis to further quantify the effect of the heave correction.

Page 11, line 230 : typo vertical Corrected

Page 12, Figure 6 shows linearized fit from sonic only, what would be the one from corrected spectrum ? Discuss values retrieved from the range of the fit identified from the residual errors and confidence in the correction.

We added the spectrum for both cloud radar Doppler velocity and sonic. Also we calculated the standard deviation of all good fits to estimate the uncertainty of the approach.

Page 12, Figure 6 legend : refers to values of EDR, but hypotheses for deriving EDR from radar should be more discussed (see above).

First, we have removed the subfigure 6(b) as suggested by Reviewer 2. Concerning the approach of EDR retrieval from radar data, we provide an extensive introduction to the topic in Section 3.2.1.

Page 12, line 251 : lacono et al., 2208, is not a general reference for RRTMG. This ref is to be replaced by a more appropriate one. We replaced the reference by Mlawer 1997, Barker 2003 Clough 2005.

Page 13, line 288 : It is OK here, but more generally for Arctic clouds I am not sure of that, as for supercooled precipitating clouds We removed the respective sentence about liquid water attenuation.

Page 14, line 310 : a strong attenuation **Done**

Page 14, line 318 : I would suggest to use scattering ratio Sr as well, which would further allow to discuss fog issue using lidar measurements only assuming a threshold in Sr

Dealing with lidar signals in the very near range (below 300m) is bound to the presence of technical caveats. Mainly, it is the incomplete overlap of the receiver-field-of-view and the laser beam. Derivation of physical values, such as SR or att. BSC, from single-channel elastically backscattered light is thus impossible very close to the ground, even for the near-range channels of PollyXT (complete overlap at 120 m). We thus decided to rely on the utilization of the technical value SNR for the detection of the cloud layers below 50 and 160 m height, which delivers very good results.

Page 15, line 348 : I would suggest to extend presentation here and discuss meteorological context change to introduce cases studies and overall meteorological patterns observed leading to the various cases analyzed. I would suggest to move Figure 11 here and briefly discuss more general transport evolution over the period studied (not necessarily adding a figure). We added an introduction to the overall synoptic situation based on Knudsen et al (2018) who gave a synoptic overview of the PS106 campaign.

Page 20, Figure 10 : I would suggest to present lidar scattering ratio instead of backscattering coefficient (to better support aerosol/fog/cloud discrimination).

The provided attenuated backscatter coefficient is a standard product of the Polly-XT processing chain (Baars et al., 2017). We thus would like to keep this parameter presented in Fig. 12 (version of Figure 10 from before the revision), as it is the default lidar parameter in Cloudnet and as we show it as standard parameter in other publications.

Page 23, and 24 : Synergies between the remote sensing instruments and auxiliary observations from aboard Polarstern were analyzed by means of Cloudnet classification procedure. This procedure is shown to induce caveats because of limitations in the radar range measurements. More discussions on the way this could be mitigated using lidar measurements should be included.

We extended the discussion of the caveats of Cloudnet and our approach to address them with the lidar measurements.

Page 23, lines 429-432 : PollyXT "Though detected fog almost continuously during the case study, : : :". How is this done ? Explain in the text how this can be translated in an additional information below 165 m in a quantitative way from scattering ratio.

As also pointed out by Reviewer #2, the terminology 'fog' is indeed inappropriate to describe what we intend to detect. We thus renamed the 'fog' flag to 'low level stratus clouds'. This better describes that we aim with our approach on detecting clouds which are (1) located above the visibility sensor of Polarstern and (2) located below the goodperformance-range of the ceilometer CL51 (deployed on Polarstern). The Figure below (Fig. 2) demonstrates this approach and the advantages. Figure 2 (a-d) present cloud parameters as derived from the CL51 ceilometer observations aboard Polarstern during the time period from 07 Jun 2017, 21 UTC to 08 Jun 2017, 09 UTC. Figure 2 (e) shows the combined Cloudnet (>165 m) and PollyXT-based (<165 m height) cloud masks and periods of fog (horizontal blue lines) as derived from the on-board visibility sensor of Polarstern (which is Figure 17 in the manuscript). Figure 2(f) shows the liquid water path as measured by the microwave radiometer HATPRO of OCEANET. The figure demonstrates nicely the situation that frequently occurred: Almost for the whole time period, CL51 shows a cloud deck, confirming that there were actually clouds present. However, the reported cloud base is continuously above 150 m height during most of the time. Even when the visibility sensor indicated fog (22:00-23:30 UTC on 7 June), the ceilometer cloud base was > 200 m. The ceilometer also reports clouds at heights, where the combined lidar + cloud radar cloud mask from Cloudnet does not show any clouds at all. This is especially visible in the time period from 05-08 UTC on 8 June. This means, that the actual cloud base must have been located lower than the lowest height of Cloudnet. And this is when the lidar data of PollyXT is of help: The threshold of SNR>40 provides a good and reasonable estimate of the actual cloud boundaries at heights

---

## Author Comment (AC3) · 15 Jun 2020

**Authors response - AMT**

**Application of the shipborne remote sensing supersite OCEANET for profiling of Arctic aerosols and clouds during Polarstern cruise PS106 – Griesche et al.**

**Response to Anonymous Referee #2 (22 January 2020)**

We would like to thank the Anonymous Referee #2 for dedicating time in order to improve the manuscript and giving help by providing us with valuable comments and suggestions. We have revised the initial submission, and hope that the manuscript is now acceptable for publication.

Our point-by-point response to the review comments is written here in bold font.

**Overall summary of major changes:**

We would like to inform the referee about the following major changes:

- **Revision of the abstract due to suggestion of Referee #2**
- **Reprocessing of the Cloudnet data with an Arctic MWR-retrieval considering the comments of both Referees #1 and #2**
- **Included a better evaluation of the capability of the motion stabilization according to comments by both Referees #1 and #2**
- **Improved the discussion of the eddy dissipation rate and fog/low-level stratus retrievals as well as Cloudnet in general considering the comments by both Referee #1 and #2**

**Detailed responses:**

The authors describe the deployment of the Oceanet remote-sensing container during a cruise to the Arctic. Right now it is not clear if the authors want to present technical development or research findings. The authors briefly describe a new motion stabilisation platform and a new data processing method for fog detection. However, they fail to provide a validation that those are working. The remainder of the paper is dedicated to case studies. The paper is of interest to the community but needs major revisions. First of all, the authors need to make up their mind if this should be a paper for AMT or ACP. There are further major items that need to be addressed before it can be considered for publication:

- [Abstract] The Abstract appears to be more of an introduction than a concise summary of the paper. Key points of the article are missing. Please rewrite the Abstract.

**We rewrote the paragraph in such a way that it provides a more concise summary.**

- [MWR retrieval] One of the major issues with this work is related to the analysis of the microwave radiometer measurements. I do not agree with the assumption that atmo- spheric conditions in the Arctic are comparable with those during winter in the Nether- lands. In the Netherlands the minimum temperature rarely reaches 0◦C; also radiative balance is not comparable. The analysis needs to be repeated with a customised Arctic retrieval. Radiosonde

data can be obtained from several research cruises in the Arctic since 1990 and are also available from research stations around the Arctic.

**Meanwhile, we reprocessed the MWR data with an retrieval that was created by University of Cologne for the location of Ny Alesund (78.9°N, 11.8°E). We mention this in the revised manuscript. The data will also be uploaded as a new version to Pangaea. Same holds for the depending Cloudnet-processed data set on Pangaea. Fig. 1 shows the correlation of the two data sets. Overall, the correlation is quite linear, especially for the IWV/PRW. Only in the low-LWP range <100 g/m², considerable relative biases can be found.**

[Figure]

Fig. 1: Comparison of LWP (a, b) and IWV/PRW (b) derived from (unflagged) MWR observations with retrievals from De Bilt and Ny Alesund.

- [Motion stabilisation] The authors should provide proof that the roll and pitch was actively levelled out for the motion-stabilised radar measurement. Please provide a time series of roll angles for the ship and radar during roughest sea and the probability distribution of radar roll angle for at least a 1 h period with greatest ship roll. Further information on the measurement conditions is needed to assess the performance of the motion stabilisation platform. What was the maximum roll angle? What was the ship's mean horizontal velocity when underway? What was the wave-induced velocity perturbation in open water?

**The discussion of the motion stabilization and its influence on the measurements has been improved. We have compared the pitch and roll movement of the Polarstern to the respective measurements of a small single board computer (Beaglebone Blue) mounted on the cloud radar during different periods of the campaign. From the comparison, we conclude that the stabilization was challenging when RV Polarstern cruised through open waters. Under these conditions, the vertical pointing accuracy could be reduced to within 1° off-zenith. While breaking through the ice and during the ice-floe period, the platform stabilized the vertical pointing with an accuracy of 0.5°.**

**The heave correction was further analyzed and quantified by investigating the Doppler velocity spectrum of the corrected and uncorrected Doppler velocity. The applied heave correction reduced the signal induced by the vertical movement of the cloud radar in the power spectral density of the Doppler velocity by a factor of 15.**

- [Eddy dissipation rate] The validation of eddy dissipation rate is not convincing. At what height was the tethered balloon located? Below a cloud or within a cloud? What are the reasons for the over- and underestimation?  Also, it would be good to have more than two comparisons cases between the Radar and the measurements with the tethered balloon or to provide justification why this is not done. Please also provide the ÉZ˙ values from tethered balloon and radar for both cases.

**We have extended the discussion on the EDR retrieval and included additional information, like the height of the tethered balloon, in the manuscript. The standard variation of the derived EDR values has been calculated to better evaluate the retrieved values. We also added another comparison of EDR between the cloud radar and the tethered balloon. Adding more comparisons is not possible as the measurement strategy of the balloon was not only focused on clouds and therefore no more co-located observations are available.**

- [Cloudnet and cloud definition] There are several issues related to the Cloudnet retrieval. Right now it is often unclear what has been done. For instance, the description of the classification mask (Page 13, line 261) does not agree with the shown Cloudnet target classification in Figure 9a. Please provide more details on Cloudnet in general  and on the classification mask and the target classification for readers that are not fa- miliar with the method. Further, it is not clear if the presented definition of liquid and mixed-phase clouds (page 12, second paragraph) is an official Cloudnet product such as the target classification or if it is a new data product developed by the authors. In that context, why not use the target classification as in comparable studies based on multi-sensor retrievals?  In those, Arctic mixed phase clouds are defined when both liquid/supercooled water and ice particles are present and when ice particles are identified directly below liquid and mixed-phase regions (e.g.  Shupe 2011, Mioche et al. 2015).  For comparison of cloud statistics from different campaigns it is important to use the same definition as already used in the literature.

**We agree with the reviewer that the Cloudnet retrieval has certain caveats. This study presents a calibrated data set of measurements, which is suitable for synergistic retrievals such as Cloudnet. To provide comparable statistics to other retrievals the data set should be processed with the respective retrieval. We adapted our introduction of Cloudnet to the simpler classification mask, which is based on the categorization bits. This classification mask is used in the manuscript in Figures 11(a) and 16(a).**

**The differentiation between supercooled liquid clouds and mixed-phase clouds at a temperature right below 0°C remains difficult. In the mentioned Figure, a cloud radar pixel was detected right below the cloud. In this situation, it is not possible with present remote sensing methods to differentiate between ice and supercooled liquid. This is only done by dew point temperature. In this case, the cloud top temperature was very close to 0°C and supercooled liquid has been found in Arctic stratiform clouds down to -4°C (Zhang et al., 2017).**

- [Fog detection] The information related to the fog detection is not adequate to evaluate if the proposed method works. Please make use of the visibility sensor aboard Polarstern to assess your findings as well as to test if your assumed SNR value of 40 can be used to reliably detect fog. Just as a reminder, fog is defined when the visibility is below 1 km. The visibility sensor can also be used to distinguish between fog and low clouds. In that regard, please compare the detected low cloud layers with the observation of the ceilometer aboard Polarstern. The first height bin is much lower than the first height bin of the Polly system. Also would it not be better to use the ceilometer for detection of fog and low cloud layers? First of all the first cloud layer is lower and the Ceilometer on Polarstern is a CL51 which reports the vertical visibility in case that the lowest height bins are obscured due to precipitation and/or fog?

**We decided to change our naming and to call our new product low-level stratus cloud instead of fog. This is closer to reality as the Polly system is only able to observe clouds starting from a height of >50m. To assess whether the SNR of 40 is a reasonable, we made a comparison between the low-level stratus cloud occurrence using three different SNR thresholds in addition to the occurrence of fog by means of the horizontal visibility sensor from Polarstern. Figure 2 shows this comparison. In blue (green) the low stratus occurrence due to a SNR of 20 (60) is indicated. Orange shows the original findings with the SNR of 40. The dashed red lines shows the frequency of occurrence of horizontal visibility below 1 km. The SNR value of 40 was manually found to provide the best visual correlation with the visibility measurements as well as to signatures of signal attenuation in the time-height cross-sections of the Cloudnet attenuated backscatter coefficient and HATPRO LWP measurements (see, e.g., Fig. 4 of this reply letter or Figs. 10 and 14 in the manuscript).**

[Figure]

Fig. 2: Comparison of low stratus occurrence due to a SNR of 20 (blue), 40 (orange) and 60 (green) and visibility below 1 km (HV, dashed red).

Minor issues

- Line 88, Please also cite Ehrlich et al. (2019) for ACLOUD
**Done**

- Line 122: It is not clear if only winter time radiosondes from De Bilt are used in the retrieval. Please clarify. But even better would be to revise the retrieval using actual Arctic measurements.
**We used a retrieval the location of Ny Alesund. See answer to major comment above.**

- Line 126:  An Arctic retrieval based on ERA-Interim data should be compared to a retrieval based on Radiosonde data.  Systematic errors in ERA-Interim data (e.g. Wesslén et al., 2014, for temperature bias) can have an influence on the MWR retrieval. Consider using ERA5 instead.
**We used a retrieval the location of Ny Alesund. See answer to major comment above.**

- line 153: Please provide the typical error range of the RS92 measurements
**Done**

- line 270: Do you mean T or Td (dew point temperature here).
**We deleted this paragraph (but Td was meant).**

- line 344: Can you please verify if the mixing depth provided by GDAS1 is comparable to the observed mixing depth. It is known that models have problems to provide realistic mixing depth in the Arctic.
**We have removed the mixing depth analysis provided by trace from the study. Nevertheless, we provide ensemble trajectories in Figure 8 in the revised manuscript. There it is well demonstrated that approximately 50% of the ensemble trajectories passed the European continent where they were involved in boundary layer processes, as indicated by the PBLH values of the underlying GDAS1 data.**

- line 354: Do you mean Figure 4.1.1? And the profiles are shown to a height of 2.5 km not 2.0 km.
**Indeed, we wanted to refer to (old) Figure 4.1.1. We changed the text also to 2.5km.**

- line 356: Since ice particles are below the liquid stratocumulus the cloud should be reclassified as mixed-phase cloud (see major comments).
**We agree that this is an ambiguous cloud situation. Still we rather stick with liquid or liquid-dominated cloud. See also answer to major comment on the Cloudnet retrieval.**

- line 375: As observed by Shupe et al. (2013). Please add citation.
**Done**

- The figures do not appear in the order they are discussed in the text. Please revise.
**Done**

- Figure 6b is not necessary and should be omitted.
**Figure 6b has been removed.**

- Figure 15: Fog and low cloud should have different colours. Again use the visibility sensor to distinguish between fog and low clouds. Add visibility to the plot. Also add the observed backscatter from the CL51 as comparison as an extra plot next to it.

**As also pointed out by Reviewer #1, the terminology 'fog' is indeed inappropriate to describe what we intend to detect. We thus renamed the 'fog' flag to 'low level stratus clouds'. This better describes that we aim with our approach on detecting clouds, which are (1) located above the visibility sensor of Polarstern and (2) located below the good-performance-range of the ceilometer CL51 (deployed on Polarstern). The Figure below (Fig. 4) demonstrates this approach and the advantages. Figure 4(a-d) present cloud parameters as derived from the CL51 ceilometer observations aboard Polarstern during the time period from 07 Jun 2017, 21 UTC to 08 Jun 2017, 09 UTC. Figure 4(e) shows the combined Cloudnet (>165 m) and PollyXT-based (<165 m height) cloud masks and periods of fog (horizontal blue lines) as derived from the on-board visibility sensor of Polarstern (which is Figure 17 in the manuscript). Figure 4(f) shows the liquid water path as measured by the microwave radiometer HATPRO of OCEANET. The figure demonstrates nicely the situation that frequently occurred: Almost for the whole time period CL51 shows a cloud deck, confirming that there were actually clouds present. However, the reported cloud base is continuously above 150 m height during most of the time. Even when the visibility sensor indicated fog (22:00-23:30 UTC on 7 June), the ceilometer cloud base was >200 m. The ceilometer also reports clouds at heights, where the combined lidar + cloud radar cloud mask from Cloudnet does not show any clouds at all. This is especially visible in the time period from 05-08 UTC on 8 June. This means, that the actual cloud base must have been located lower than the lowest height of Cloudnet. And this is when the lidar data of PollyXT is of help: The threshold of SNR>40 provides a good and reasonable estimate of the actual cloud boundaries at heights <165 m .**

**We decided to not do a detailed discussion of the issues of the CL51 within the manuscript. However, from our observations it is clear, that the reported cloud bases from the CL51 are continuously too high, at least in situations with very low clouds present. We hope that Figure 4 demonstrates well to the reviewers that the new cloud mask from PollyXT is valuable. The cloud mask will also be published in Pangaea to provide other users a good estimate of the low-cloud occurrence - a very important parameter for the radiative and water balances.**

- Figure 17: How is fog height determined? That needs to be discussed in 3.3.3. Add visibility to the plot.
**The low level stratus height was determined by the lowest and the highest PollyXT pixel of the low level stratus mask, which exceeded the SNR-threshold. We added a flag to indicate periods of horizontal visibility < 1 km in Figure 17.**

- line 500, e.g. Sotiropoulou et al. (2014) and (2016) considered low clouds from ceilometer/Halo and radar measurements.
**We have considered their findings in our discussion.**

- Line 762: Somag, the provided link does not work. Please provide an open link or add the information to the text.
**Done**

- Figures 7, 10 (upper panel), and 13: Please use same scale for T and RH in all plots.
**Done**

[Figure]

Fig. 4: Comparison of ceilometer-derived cloud bases, PollyXT 'low stratus' detection mask and visibility sensor for the Polarstern observations from 07 Jun 2017, 21 UTC to 08 Jun 2017, 09 UTC. (a) Detection Status of the CL51 Ceilometer; (b) vertical visibility from CL51 (if detection status equals 4); (c) and (d) height of lowest cloud base from CL51; (d) Cloudnet cloud and aerosol mask (above 165 m), PollyXT low-stratus mask (below 165 m) and fog-periods (horiz.

blue lines) as derived from the visibility sensor; (f) liquid water path as derived from the microwave radiometer HATPRO.

References:

Ehrlich, A., Wendisch, M., Lüpkes, C., Buschmann, M., Bozem, H., Chechin, D., . . . Zanatta, M. (2019). A comprehensive in situ and remote sensing data set from the Arctic CLoud Observations Using airborne measurements during po- lar Day (ACLOUD) campaign. Earth System Science Data Discussions, 1–42. https://doi.org/10.5194/essd-2019-96

Wesslén, C., Tjernström, M., Bromwich, D. H., De Boer, G., Ekman, A. M. L., Bai, L. S., & Wang, S. H. (2014). The Arctic summer atmosphere: An evaluation of re- analyses using ASCOS data. Atmospheric Chemistry and Physics, 14(5), 2605–2624. https://doi.org/10.5194/acp-14-2605-2014

Shupe, M. D.: Clouds at Arctic Atmospheric Observatories, Part II: Ther- modynamic phase characteristics, J. Appl. Meteor. Clim., 50, 645–661, doi:10.1175/2010JAMC2468.1, 2011.

Mioche, G., Jourdan, O., Ceccaldi, M., and Delanoë, J.: Variability of mixed- phase clouds in the Arctic with a focus on the Svalbard region: a study based on spaceborne active remote sensing, Atmos. Chem. Phys., 15, 2445–2461, https://doi.org/10.5194/acp-15-2445-2015, 2015.

Shupe, M. D., Persson, P. O. G., Brooks, I. M., Tjernström, M.,Sedlar, J., Mauritsen, T., Sjogren, S., and Leck, C.: Cloud and boundary layer interactions over the Arctic sea ice in late sum-mer, Atmos. Chem. Phys., 13, 9379–9399, doi:10.5194/acp-13-9379-2013, 2013.

Sotiropoulou, G., Sedlar, J., Tjernström, M., Shupe, M. D., Brooks, I. M., and Pers- son, P. O. G.: The thermodynamic structure of summer Arctic stratocumulus and the dynamic coupling to the surface, Atmos. Chem. Phys., 14, 12573–12592, https://doi.org/10.5194/acp-14-12573-2014, 2014.

Sotiropoulou, G., Tjernström, M., Sedlar, J., Achtert, P., Brooks, B. J., Brooks, I. M.,

Wolfe, D. (2016). Atmospheric conditions during the Arctic clouds in summer experi- ment (ACSE): Contrasting open water and sea ice surfaces during melt and freeze-up seasons. Journal of Climate, 29, 8721-8744

---

## Author Response (AR2)

**Authors response - AMT**

**Application of the shipborne remote sensing supersite OCEANET for profiling of Arctic aerosols and clouds during Polarstern cruise PS106 – Griesche et al.**

Dear Editor,

Many thanks for your feedback. The upload of the required files is currently under progress.

While evaluating the CL51 data we indeed mixed feet with meter. Scaling the data to the proper unit improved the agreement between both measurement. As we have not used any ceilometer data in the manuscript, we think this does not affect the message. But this encourages us anyway to incorporate the CL51 data in our future studies.

We have implemented your edits to the text and updated the mentioned citation (line 742) as the upload of the cited data set is now finalized and available.

In addition, we added a minor correction to our heave rate calculation. So far, we have not considered the displacement of the vertical axis of the velocity vector resulting from the rotation of the RV (result of Eq. (1)) because of the rather central position of the cloud radar aboard Polarstern. We have now incorporated this by considering the rotation matrix (Eq. (2) in the new manuscript). As can be seen, given the small pitch and roll angles of Polarstern during the cruise (Fig. 3) in the order of 0 to 5°, the effect of the new term on the z-movement is in the order of < 1%.

Also we added Andreas Macke as another Coauthor of the manuscript.

Please find attached the latest marked-up manuscript version.

Best regards,

Hannes Griesche

[revised manuscript text omitted]